# Association between thyroid hormone sensitivity and ischemic stroke-associated pneumonia: The role of FT3/FT4 ratio

Zhidan Hua[1], Zichen Rao [2]*, Yiming Zhang[2], Chunyan Zhu[2]

**1** Department of Pulmonary and Critical Care Medicine, The Quzhou Affiliated Hospital of Wenzhou Medical University, Quzhou People's Hospital, Quzhou, Zhejiang, China, **2** Department of Endocrinology, The Quzhou Affiliated Hospital of Wenzhou Medical University, Quzhou People's Hospital, Quzhou, Zhejiang, China

* rzc1522@wmu.edu.cn

## Abstract

### Background

Thyroid hormone sensitivity has emerged as a critical factor in various diseases. However, its relationship with ischemic stroke-associated pneumonia (iSAP) in euthyroid patients with ischemic stroke remains poorly understood. This study aims to elucidate the association between thyroid hormone sensitivity indices and iSAP risk.

### Methods

A total of 1,767 euthyroid patients with ischemic stroke were enrolled and categorized into the iSAP group (n = 376) and the non-iSAP group (n = 1,391). Univariate and multivariate logistic regression analyses were performed to assess the association between thyroid hormone sensitivity indices-including free triiodothyronine (FT3), free thyroxine (FT4), FT3/FT4 ratio, thyroid-stimulating hormone index (TSHI), Thyrotroph T4 Resistance Index (TT4RI), and thyroid feedback quantile-based indices (TFQI-FT3, TFQI-FT4)-and iSAP risk. The predictive performance of these indices was evaluated using receiver operating characteristic (ROC) curve analysis.

### Results

Compared with the non-iSAP group, patients with iSAP were older and exhibited a higher prevalence of atrial fibrillation (AF) and chronic obstructive pulmonary disease (COPD), along with greater stroke severity (higher NIHSS scores). Univariate analysis demonstrated that higher FT3 levels (OR = 0.85, 95% CI: 0.80–0.91, p < 0.0001) and a higher FT3/FT4 ratio (OR = 0.80, 95% CI: 0.69–0.92, p = 0.0018) were statistically associated with lower odds of developing iSAP. After adjusting for confounders, multivariate analysis revealed that a higher FT3/FT4 ratio remained inversely associated with iSAP occurrence (Q3 vs. Q1: OR = 0.40, 95% CI: 0.26–0.62, p < 0.0001; Q4

**Data availability statement:** Data cannot be shared publicly because of patient privacy and confidentiality restrictions. Data are available from the Quzhou Affiliated Hospital of Wenzhou Medical University Institutional Data Access / Ethics Committee (contact via ethics@qzah-wmu.edu.cn) for researchers who meet the criteria for access to confidential data.

**Funding:** The author(s) received no specific funding for this work.

**Competing interests:** The authors have declared that no competing interests exist.

vs. Q1: OR = 0.31, 95% CI: 0.19–0.48, p < 0.0001). Additionally, elevated TFQI-FT3 levels showed a significant inverse association with iSAP occurrence (OR = 0.35, 95% CI: 0.18–0.67, p = 0.0017). ROC analysis demonstrated that the FT3/FT4 ratio and the Age, Atrial fibrillation, Dysphagia, Sex, and Stroke Severity(A2DS2)score exhibited moderate predictive accuracy for iSAP, with area under the curve (AUC) values of 0.711 and 0.763, respectively.

## Conclusion

In euthyroid patients with ischemic stroke, a lower FT3/FT4 ratio and reduced TFQI-FT3 levels were linked to higher odds of iSAP. These exploratory findings suggest that thyroid hormone sensitivity indices, particularly the FT3/FT4 ratio, may serve as potential predictive markers and warrant validation in prospective studies.

## Introduction

Ischemic stroke remains a leading cause of morbidity and mortality worldwide, with its incidence rising steadily, imposing a substantial burden on both patients and healthcare systems [1–3]. Among post-stroke complications, ischemic stroke-associated pneumonia (iSAP) is a critical determinant of poor clinical outcomes [4,5]. iSAP not only prolongs hospitalization and escalates healthcare costs but also markedly increases mortality and disability rates [4,6]. Emerging evidence suggests that iSAP is closely linked to post-stroke dysphagia, immune dysregulation, and neuroendocrine disturbances [7]. Consequently, early identification of high-risk patients and timely intervention are essential for improving prognosis and reducing complications.

Thyroid hormones play a pivotal role in regulating metabolism, cardiovascular function, and neurological homeostasis [8,9]. In recent years, thyroid hormone sensitivity has garnered attention as a potential factor influencing various diseases, particularly cardiovascular disorders and metabolic syndromes [9]. However, its role in neurological disorders, including ischemic stroke and its complications, remains insufficiently explored. While previous studies have established an association between thyroid dysfunction and increased stroke risk [10], it remains unclear whether thyroid hormone sensitivity is independently linked to iSAP in euthyroid patients with ischemic stroke.

Thyroid hormones may influence the development of iSAP by regulating metabolism, immune responses, and inflammation [11]. FT3 and FT4, the active thyroid hormones, are typically measured as a ratio (FT3/FT4), reflecting peripheral hormone conversion efficiency. Additionally, thyroid hormone sensitivity indices, such as the thyroid feedback quantile index (TFQI) and the TT4RI, provide a more nuanced assessment of thyroid hormone bioactivity [12,13]. Despite their potential significance, limited research has examined the association between these indices and iSAP, particularly in euthyroid stroke patients.

This study aims to elucidate the relationship between thyroid hormone sensitivity indices and iSAP risk, as well as to evaluate their predictive value in ischemic stroke

patients. By enhancing our understanding of this association, the findings may contribute to the development of novel risk stratification strategies and targeted interventions for iSAP prevention.

## Materials and methods

### Study design and participants

The data for this study were derived from a previously established retrospective cohort study on the prognosis of ischemic stroke.

Although this dataset has supported prior publications from our team on different research questions, the current study is the first to investigate the association between thyroid hormone sensitivity indices (e.g., FT3/FT4 ratio, TFQI, TSHI, TT4RI) and iSAP risk. No part of the present analysis has been published previously.

Patient data meeting the inclusion criteria from September 2016 to September 2022 at our hospital were collected. All included patients were enrolled into a single-center retrospective cohort and were subsequently stratified into two groups based on whether they developed ischemic stroke-associated pneumonia (iSAP) during hospitalization: an iSAP group and a non-iSAP group. Pre-admission infection and antibiotic use status were assessed retrospectively based on our hospital's integrated electronic medical records, including outpatient visit notes, admission assessments, medication orders, and nursing documentation. This system also incorporates provincial-level patient visit records, which are reviewed and summarized by admitting physicians. Thyroid hormone-related indicators were measured at the time of admission, and the occurrence of iSAP was subsequently monitored throughout the hospital stay. This was not a case-control study, and no matching based on case-control design was used; group differences were addressed through multivariate adjustment. This study was approved by the Ethics Committee of Quzhou People's Hospital (Approval Number: 2023−151), which granted a waiver of informed consent due to the retrospective nature of the study and the use of de-identified data. The data were accessed for analysis on Oct 16, 2024. Prior to analysis, all data were anonymized, and researchers did not have access to any personally identifiable information.

Inclusion criteria were as follows: (1) confirmed diagnosis of AIS within 24 hours of onset with head magnetic resonance imaging (MRI) completed within 48 hours; (2) age ≥ 18 years;

Exclusion criteria included: (1) patients in the non-acute or recovery phase of stroke; (2) pregnant women; (3) patients with thyroid disorders or receiving thyroid medications; (4) Patients who had infectious diseases or fever within 2 weeks prior to admission and those who used antibiotics within 1 week before admission were excluded.

The detailed patient selection process is illustrated in Fig 1.

### Baseline data collection

Baseline data were retrieved from medical records and included demographic and clinical characteristics at admission, such as age, sex, smoking status, hypertension, type 2 diabetes, atrial fibrillation (AF) and chronic obstructive pulmonary disease (COPD). COPD diagnoses were based on confirmed prior clinical diagnosis as documented in the medical records, rather than self-reported symptoms or medication use. Stroke severity was assessed using the National Institutes of Health Stroke Scale (NIHSS), swallowing difficulty was evaluated using the Kubota Water Drinking Test (KWDT), and consciousness disturbances were assessed using the Glasgow Coma Scale (GCS).

Blood samples were collected by trained nurses using vacuum tubes on the second morning of admission (6:00 AM), stored at 4°C, and analyzed within 2 hours by certified clinical laboratory technicians. Measured parameters included white blood cells (WBC), C-reactive protein (CRP), fasting plasma glucose (FPG), aspartate aminotransferase (AST), alanine aminotransferase (ALT), glycated hemoglobin (HbA1c), homocysteine (HCY), serum creatinine (SCr, measured from serum samples), albumin (ALB), triglycerides (TG), total cholesterol (TC), high-density lipoprotein cholesterol (HDL-c), and low-density lipoprotein cholesterol (LDL-c). Additionally, FT4, FT3, and TSH levels were measured using

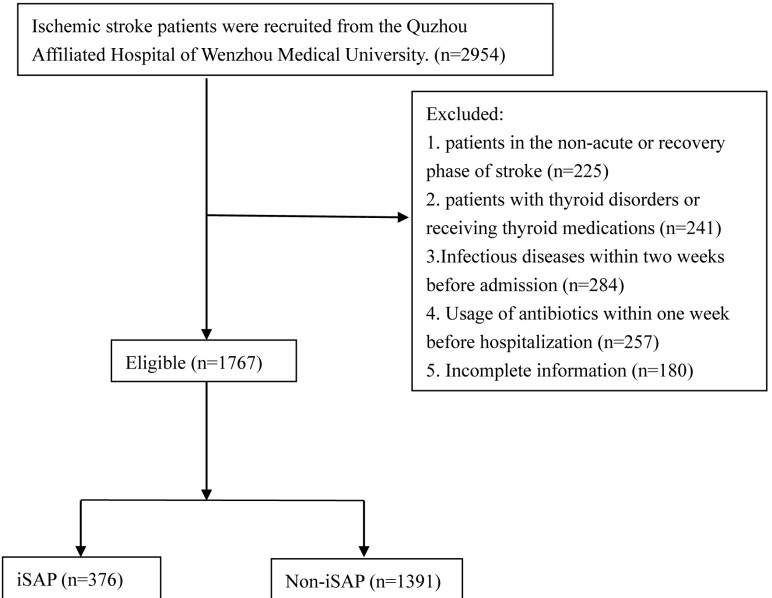

**Fig 1. Patient Selection Flowchart for the Study of FT3/FT4 Ratio and Ischemic Stroke-Associated Pneumonia (iSAP).** A total of 2,954 ischemic stroke patients were initially screened at the Quzhou Affiliated Hospital of Wenzhou Medical University. Patients were excluded if they were in the non-acute or recovery phase of stroke (n = 225), had thyroid disorders or were receiving thyroid-related medications (n = 241), had infectious diseases within two weeks before admission (n = 284), had used antibiotics within one week before hospitalization (n = 257), or had incomplete information (n = 180). Finally, 1,767 eligible patients were included and stratified into two groups according to the development of ischemic stroke-associated pneumonia (iSAP) during hospitalization: the iSAP group (n = 376) and the non-iSAP group (n = 1,391).

a chemiluminescent immunoassay (CLIA), a standardized and widely accepted method endorsed by major international endocrine societies. The estimated glomerular filtration rate (eGFR) was calculated using the CKD-EPI formula to assess renal function, with a normal reference range of 90–120 mL/min/1.73 m². The normal reference ranges were: TSH, 0.27–4.2 mIU/L; FT4, 12–22 pmol/L; FT3, 3.1–6.8 pmol/L.

## Definitions

To ensure iSAP was acquired during hospitalization, all patients underwent routine chest CT imaging and clinical infection screening (including temperature, WBC, CRP, and physical examination) at admission. Those presenting radiographic or clinical evidence of active infection at baseline were excluded from analysis. The diagnosis of SAP was determined independently by two attending neurologists following the Pneumonia in Stroke Consensus Group recommendations. If necessary, an attending respiratory physician was consulted for confirmation.

The diagnostic criteria required at least one of the following:

Fever (>38°C) without an alternative cause;

Abnormal WBC count (leukopenia <4 × 10⁹/L or leukocytosis >12 × 10⁹/L);

Altered mental status in patients ≥70 years without other causes.

Plus at least two of:

Purulent sputum or a change in sputum character;

Increased respiratory secretions or suction needs;

New or worsening cough, dyspnea, or tachypnea (>25 breaths/min);

Auscultatory findings of rales, crackles, or bronchial breath sounds;

Oxygen desaturation ($PaO_2/FiO_2 \le 240$) or increased oxygen requirement;

Radiologic confirmation was required with two consecutive chest X-rays showing new or progressive infiltrates, consolidation, or cavitation. In patients without prior pulmonary or cardiac disease, a single conclusive chest radiograph was deemed sufficient. Given the study's focus on ischemic stroke-associated pneumonia, chest CT imaging was utilized instead of chest X-rays, providing superior diagnostic clarity. These criteria were selected for their relevance to the study population and their validated effectiveness in pneumonia diagnosis among stroke patients.

The formulas for thyroid hormone (TH) sensitivity indices were as follows:

1. FT3/FT4: FT3 (pmol/L)/ FT4 (pmol/L), reflecting peripheral TH sensitivity and the efficiency of FT4-to-FT3 conversion. Higher ratios indicate increased peripheral TH sensitivity.

2. TT4RI: FT4 (pmol/L) × TSH (mIU/L), associating TSH with FT4. Higher TT4RI values suggest lower central TH sensitivity.

3. TSHI: lnTSH (mIU/L) + 0.1345×FT4 (pmol/L), representing feedback inhibition of FT4 on TSH. Higher TSHI values reflect reduced central TH sensitivity.

4. TFQI FT4: cdfFT4 − (1−cdfTSH). Values range from −1–1, with negative values indicating increased central TH sensitivity and positive values indicating decreased sensitivity.

5. TFQI FT3: cdfFT3 − (1−cdfTSH). Values range from −1–1, with negative values indicating increased central TH sensitivity and positive values indicating decreased sensitivity.

Because thyroid hormone sensitivity indices such as FT3/FT4 ratio, TT4RI, TSHI, TFQI FT4 and TFQI FT3 lack universally accepted clinical reference ranges, these variables were categorized into quartiles (Q1-Q4) based on their distribution within our study population for comparative analysis.

## Statistical analysis

Statistical analyses were conducted using R Studio (version 4.2.2; R Foundation for Statistical Computing, Vienna, Austria) and EmpowerStats (version 2.0; www.empowerstats.com). Continuous variables were assessed for normality with the Kolmogorov–Smirnov test. Normally distributed variables were reported as means±SD and compared using Student's t-tests, while non-normally distributed variables were reported as medians (IQRs) and compared using the Mann–Whitney U test. Categorical variables were compared using Chi-square or Fisher's exact tests.

To reduce potential confounding caused by imbalances in baseline characteristics, we applied propensity score matching (PSM) combined with Genetic Matching (GenMatch) algorithms. Matching was performed in a 1:1 ratio using nearest-neighbor methods with replacement, allowing some iSAP patients to be matched multiple times to maximize covariate balance. The final matched cohort included 2926 patients (1463 iSAP and 1463 non-iSAP). This approach has been widely adopted in observational studies for its capacity to enhance covariate balance, especially in scenarios with baseline group imbalances. Following the completion of matching, we conducted multivariate logistic regression analyses based on the matched dataset.

Univariate logistic regression analyses were first performed to screen for potential predictors of ischemic stroke-associated pneumonia (iSAP). Variables with a p-value<0.1 in the univariate analysis were subsequently included in multivariate logistic regression models using a backward stepwise elimination approach. Three models were constructed: (Model 1) unadjusted (crude); (Model 2) adjusted for age and sex; and (Model 3) fully adjusted for demographic and

clinical variables, including age, sex, smoking status, hypertension, diabetes mellitus, atrial fibrillation (AF), chronic obstructive pulmonary disease (COPD), fasting plasma glucose (FPG), high-density lipoprotein cholesterol (HDL-C), low-density lipoprotein cholesterol (LDL-C), C-reactive protein (CRP), blood urea nitrogen (BUN), uric acid (UA), homocysteine (HCY), white blood cell count (WBC), estimated glomerular filtration rate (eGFR), glycated hemoglobin A1c (HbA1c), triglycerides (TG), and aspartate aminotransferase (AST). Odds ratios (ORs) and 95% confidence intervals (CIs) were reported. A two-tailed p-value < 0.05 was considered statistically significant.

Receiver operating characteristic (ROC) curve analysis was conducted to evaluate the predictive performance of the FT3/FT4 ratio and the $A_2DS_2$ score in identifying patients at risk for iSAP. The AUC for each model was calculated and compared using the DeLong test.

## Results

### Baseline characteristics

The baseline characteristics of patients with and without ischemic stroke-associated pneumonia (iSAP) are summarized in Table 1. Patients in the iSAP group were significantly older (median: 77.00 vs. 69.00 years, p < 0.001) and exhibited a higher prevalence of atrial fibrillation (35.11% vs. 11.50%, p < 0.001), chronic obstructive pulmonary disease (COPD) (15.69% vs. 4.03%, p < 0.001), and coronary heart disease (23.35% vs. 14.81%, p = 0.002). The Glasgow Coma Scale (GCS) scores were notably lower in the iSAP group (median: 10.00 vs. 14.00, p < 0.001), indicating impaired consciousness.

Swallowing dysfunction, as assessed by the Kubota Water Drinking Test (KWDT), was significantly more severe in the iSAP group, with a greater proportion of patients exhibiting grade III or IV swallowing impairment (61.78% vs. 15.80%, p < 0.001). Additionally, the National Institutes of Health Stroke Scale (NIHSS) scores were significantly higher at admission in the iSAP group (median: 11.00 vs. 6.00, p < 0.001), reflecting greater stroke severity. Detailed data are shown in Table 1.

### Thyroid hormone profiles

Patients with iSAP exhibited notable alterations in thyroid hormone levels. Free triiodothyronine (FT3) levels were significantly lower (median: 3.44 vs. 4.09 pmol/L, p < 0.001), while the FT3/FT4 ratio was also reduced (0.21–0.30 vs. 0.26–0.38, p < 0.001), suggesting impaired peripheral thyroid hormone conversion. Conversely, free thyroxine (FT4) levels were elevated (median: 13.50 vs. 12.56 pmol/L, p < 0.001), whereas thyroid-stimulating hormone (TSH) levels were lower (median: 1.33 vs. 1.72 mIU/mL, p < 0.001). Detailed data are presented in Table 1.

Thyroid hormone sensitivity indices also differed significantly between the groups. TSHI, TFQI-FT3, and TFQI-FT4 were markedly lower in the iSAP group, while TT4RI was significantly elevated (p < 0.001). These findings suggest that dysregulation of thyroid hormone metabolism, particularly reduced FT3 levels and altered hormone sensitivity, may contribute to the pathophysiology of iSAP. Corresponding sensitivity indices are also detailed in Table 1.

To address the imbalance in baseline characteristics between the iSAP and non-iSAP groups in the original dataset, we applied propensity score matching (PSM) integrated with the Genetic Matching algorithm to construct a well-balanced matched cohort. After matching, a total of 2,926 patients (1,463 in each group) were included. As shown in S1 Table and S1 Fig, the absolute standardized mean differences (SMDs) for most baseline covariates were substantially reduced and fell below the recommended threshold of 0.1, indicating excellent covariate balance between the two groups. This methodological improvement significantly enhanced the comparability of the matched groups and reduced the risk of confounding bias in subsequent regression analyses.

### Association between thyroid hormone sensitivity and iSAP risk

Several significant risk factors for iSAP were identified through univariate logistic regression analysis in Table 2, including advanced age (OR = 1.05, 95% CI: 1.04–1.06, p < 0.0001), atrial fibrillation (OR = 4.16, 95% CI: 3.18–5.44, p < 0.0001),

**Table 1. Baseline characteristics of patients with iSAP and non-iSAP.**

| Characteristic | non-iSAP (n = 1391) | iSAP (n = 376) | p-value* |
|---|---|---|---|
| AGE (years) | 69.00 (60.00-77.00) | 77.00 (67.00-83.00) | <0.001 |
| Male, n (%) | 840 (60.39%) | 211 (56.12%) | 0.134 |
| Current smoking, n (%) | 511 (36.74%) | 138 (36.70%) | 0.990 |
| Hypertension, n (%) | 1071 (76.99%) | 287 (76.33%) | 0.786 |
| Diabetes, n (%) | 497 (35.73%) | 131 (34.84%) | 0.749 |
| Atrial fibrillation, n (%) | 160 (11.50%) | 132 (35.11%) | <0.001 |
| COPD, n (%) | 56 (4.03%) | 59 (15.69%) | <0.001 |
| SBP (mmHg) | 152.21 ± 21.59 | 152.46 ± 20.96 | 0.807 |
| DBP (mmHg) | 82.88 ± 12.98 | 81.76 ± 12.78 | 0.239 |
| FPG (mg/dL) | 115.43 ± 43.97 | 126.96 ± 60.29 | <0.001 |
| TG (mg/dL) | 117.80 (84.14-163.85) | 93.00 (69.08-127.76) | <0.001 |
| HbA1C (%) | 6.56 ± 1.72 | 6.67 ± 1.78 | 0.153 |
| TC (mmol/L) | 4.32 (3.69-5.03) | 4.18 (3.57-4.92) | 0.022 |
| HDL-C (mmol/L) | 1.16 ± 0.30 | 1.20 ± 0.35 | 0.029 |
| LDL-C (mmol/L) | 2.87 ± 0.99 | 2.80 ± 1.04 | 0.128 |
| CRP(mg/L) | 2.00 (1.00-4.25) | 13.62 (4.00-37.47) | <0.001 |
| AST(U/L) | 21.37 ± 10.64 | 26.22 ± 18.11 | <0.001 |
| ALT(U/L) | 21.46 ± 16.48 | 22.71 ± 22.94 | 0.617 |
| BUN(mg/dL) | 5.10 (4.11-6.26) | 5.70 (4.40-7.30) | <0.001 |
| UA (umol/L) | 313.00 (257.40-382.05) | 303.75 (235.45-369.80) | 0.034 |
| WBC(×109/L) | 6.60 (5.40-8.00) | 9.24 (7.44-11.39) | <0.001 |
| EGFR (ml/min/1.73 m2) | 100.69 (81.77-119.34) | 93.86 (71.75-116.24) | <0.001 |
| Homocysteine (mmol/L) | 14.60 (11.50-19.00) | 16.30 (12.93-20.02) | <0.001 |
| FT3 (pmol/L) | 4.09 (3.55-4.70) | 3.44 (2.91-4.07) | <0.001 |
| FT4 (pmol/L) | 12.56 (10.81-14.27) | 13.50 (11.92-15.04) | <0.001 |
| TSH (mIU/mL) | 1.72 (1.06-2.70) | 1.33 (0.73-2.39) | <0.001 |
| T3 (nmol/L) | 1.41 (1.26-1.54) | 1.22 (1.01-1.43) | <0.001 |
| T4 (nmol/L) | 89.80 (80.19-98.82) | 86.90 (75.92-98.16) | 0.001 |
| TSHI | 2.23 (1.62-2.75) | 2.06 (1.43-2.70) | 0.012 |
| TT4RI | 20.98 (11.70-33.56) | 17.27 (8.83-31.65) | 0.001 |
| FT3/FT4 | 0.31 (0.26-0.38) | 0.25 (0.21-0.30) | <0.001 |
| TFQI-FT4 | −0.02 (−0.17-0.14) | 0.01 (−0.13-0.15) | 0.027 |
| TFQI-FT3 | −0.13 (−0.22-0.03) | −0.23 (−0.34--0.09) | <0.001 |
| NHISS | 2.00 (1.00-5.00) | 5.00 (2.00-13.00) | <0.001 |
| KWDT | 1.00 (1.00-1.00) | 1.00 (1.00-4.00) | <0.001 |
| GCS | 15.00 (15.00-15.00) | 14.00 (12.00-15.00) | <0.001 |
| $A_2DS_2$ | 2.00 (2.00-4.00) | 4.00 (3.00-7.00) | <0.001 |

The table shows that age and thyroid hormone levels were significantly associated with iSAP risk, indicating a potential link between thyroid dysfunction and pneumonia risk after ischemic stroke.

Data are presented as mean ± standard deviation (SD), median (interquartile range), or number (%), as appropriate. Between-group comparisons were conducted using Student's t-test or Mann–Whitney U test for continuous variables and Chi-square or Fisher's exact test for categorical variables, with statistical significance indicated by p-values.

iSAP: ischemic stroke-associated pneumonia; SBP: systolic blood pressure; DBP: diastolic blood pressure; FPG: fasting plasma glucose; TG: triglycerides; HbA1C: hemoglobin A1c; TC: total cholesterol; HDL-C: high-density lipoprotein cholesterol; LDL-C: low-density lipoprotein cholesterol; CRP: C-reactive protein; AST: aspartate aminotransferase; ALT: alanine aminotransferase; BUN: blood urea nitrogen; UA: uric acid; WBC: white blood cell count; eGFR: estimated glomerular filtration rate; FT3: free triiodothyronine; FT4: free thyroxine; TSH: thyroid-stimulating hormone; TSHI, thyroid-stimulating hormone index; TT4RI, thyrotroph T4 resistance index; TFQI-FT3, thyroid feedback quantile-based index calculated with FT3; TFQI-FT4, thyroid feedback quantile-based index calculated with FT4; NIHSS: National Institutes of Health Stroke Scale; KWDT: Kubota Water Drinking Test; GCS: Glasgow Coma Scale; $A_2DS_2$: Age, Atrial Fibrillation, Dysphagia, Sex, and Stroke Severity Score.

Reference ranges: FT3 (3.5–6.5 pmol/L), FT4 (10.0–22.0 pmol/L), TSH (0.27–4.20 mIU/L), T3 (1.1–2.9 nmol/L), T4 (60–150 nmol/L), based on standard laboratory values consistent with international guidelines (e.g., ATA, WHO).

**Table 2. Univariate analysis of Risk Factors for Ischemic Stroke-Associated Pneumonia (iSAP).**

| | Statistics | OR (95%CI) | p-value* |
|---|---|---|---|
| AGE (years) | 69.27 ± 12.45 | 1.05 (1.04, 1.06) | <0.0001 |
| Gender, n (%) | | | |
| Female | 716 (40.52%) | 1.0 | |
| Male | 1051 (59.48%) | 0.84 (0.67, 1.06) | 0.1347 |
| Current smoking, n (%) | | | |
| No | 1118 (63.27%) | 1.0 | |
| Yes | 649 (36.73%) | 1.00 (0.79, 1.26) | 0.9903 |
| Hypertension, n (%) | | | |
| No | 409 (23.15%) | 1.0 | |
| Yes | 1358 (76.85%) | 0.96 (0.74, 1.26) | 0.7861 |
| Diabetes, n (%) | | | |
| No | 1139 (64.46%) | 1.0 | |
| Yes | 628 (35.54%) | 0.96 (0.76, 1.22) | 0.7492 |
| Atrial fibrillation, n (%) | | | |
| No | 1475 (83.47%) | 1.0 | |
| Yes | 292 (16.53%) | 4.16 (3.18, 5.44) | <0.0001 |
| COPD, n (%) | | | |
| No | 1652 (93.49%) | 1.0 | |
| Yes | 115 (6.51%) | 4.44 (3.02, 6.52) | <0.0001 |
| FPG (mg/dL) | 117.88 ± 48.12 | 1.00 (1.00, 1.01) | <0.0001 |
| TG (mg/dL) | 133.60 ± 89.30 | 0.99 (0.99, 1.00) | <0.0001 |
| HbA1C (%) | 6.59 ± 1.73 | 1.04 (0.97, 1.10) | 0.2768 |
| TC (mmol/L) | 4.38 ± 1.10 | 0.90 (0.81, 1.00) | 0.0486 |
| HDL-C (mmol/L) | 1.17 ± 0.32 | 1.48 (1.05, 2.08) | 0.0269 |
| LDL-C (mmol/L) | 2.85 ± 1.00 | 0.93 (0.83, 1.04) | 0.2139 |
| CRP (mg/L) | 10.33 ± 27.57 | 1.06 (1.05, 1.07) | <0.0001 |
| AST (U/L) | 22.40 ± 12.75 | 1.03 (1.02, 1.03) | <0.0001 |
| ALT (U/L) | 21.72 ± 18.05 | 1.00 (1.00, 1.01) | 0.2360 |
| BUN (mg/dL) | 5.61 ± 2.37 | 1.18 (1.13, 1.24) | <0.0001 |
| UA (umol/L) | 321.45 ± 97.94 | 1.00 (1.00, 1.00) | 0.0940 |
| Homocysteine (mmol/L) | 16.71 ± 9.26 | 1.01 (1.00, 1.03) | 0.0147 |
| FT3 (pmol/L) | 4.84 ± 3.20 | 0.85 (0.80, 0.91) | <0.0001 |
| FT4 (pmol/L) | 12.52 ± 5.09 | 1.03 (1.01, 1.06) | 0.0156 |
| TSH (mIU/mL) | 2.32 ± 4.01 | 0.99 (0.96, 1.02) | 0.5794 |
| TSHI | 2.14 ± 1.05 | 0.84 (0.75, 0.95) | 0.0042 |
| TT4RI | 26.99 ± 31.14 | 1.00 (0.99, 1.00) | 0.2514 |
| TFQI-FT4 | −0.02 ± 0.25 | 1.74 (1.08, 2.78) | 0.0216 |
| FT3/FT4 | 0.62 ± 1.02 | 0.80 (0.69, 0.92) | 0.0018 |
| TFQI-FT3 | −0.08 ± 0.25 | 0.08 (0.04, 0.15) | <0.0001 |
| T3 (nmol/L) | 1.37 ± 0.30 | 0.10 (0.07, 0.16) | <0.0001 |
| T4 (nmol/L) | 89.19 ± 17.51 | 0.99 (0.98, 1.00) | 0.0048 |
| WBC (×109/L) | 7.49 ± 2.81 | 1.51 (1.43, 1.59) | <0.0001 |
| EGFR (ml/min/1.73 m2) | 100.82 ± 34.34 | 0.99 (0.99, 1.00) | 0.0001 |
| NHISS | 4.36 ± 4.98 | 1.17 (1.14, 1.20) | <0.0001 |
| KWDT | 1.43 ± 1.06 | 2.54 (2.27, 2.85) | <0.0001 |

*(Continued)*

**Table 2.** (Continued)

| | Statistics | OR (95%CI) | p-value* |
|---|---|---|---|
| GCS | 14.09 ± 1.98 | 0.66 (0.62, 0.71) | <0.0001 |
| A$_2$DS$_2$ | 3.12 ± 1.85 | 1.78 (1.66, 1.90) | <0.0001 |

This table presents the statistical results of univariate logistic regression analyses examining associations between demographic, clinical, and laboratory parameters and the risk of ischemic stroke-associated pneumonia (iSAP). Baseline characteristics were analyzed using univariate logistic regression to assess their association with the risk of iSAP. Data are presented as odds ratios (ORs) with 95% confidence intervals (CIs) and corresponding p-values.

COPD (OR = 4.44, 95% CI: 3.02–6.52, p < 0.0001), and NIHSS scores (OR = 1.17, 95% CI: 1.14–1.20, p < 0.0001). Regarding thyroid hormone parameters, higher FT3 levels were associated with lower odds of developing iSAP (OR = 0.85, 95% CI: 0.80–0.91, p < 0.0001), as was a higher FT3/FT4 ratio (OR = 0.80, 95% CI: 0.69–0.92, p = 0.0018). These findings suggest that each unit decrease in FT3 or FT3/FT4 ratio corresponds to a 15% and 20% increase in the odds of iSAP, respectively.

Abbreviations are as defined in Table 1. In multivariate logistic regression analysis, the results are presented in Table 3, a higher FT3/FT4 ratio was independently associated with a lower risk of iSAP (Q3 vs. Q1: OR = 0.40, 95% CI: 0.26–0.62, p < 0.0001; Q4 vs. Q1: OR = 0.31, 95% CI: 0.19–0.48, p < 0.0001). TFQI-FT3 levels also demonstrated a consistent protective association (OR = 0.35, 95% CI: 0.18–0.67, p = 0.0017), reinforcing the potential role of thyroid hormone sensitivity in iSAP risk stratification.

Reinforcing this finding, results from the matched cohort (Table 4) showed that the FT3/FT4 ratio remained inversely associated with iSAP risk (Q2 vs. Q1: OR = 0.69, 95% CI: 0.56–0.86, p = 0.0011; Q4 vs. Q1: OR = 0.72, 95% CI: 0.57–0.91, p = 0.0054). Additionally, TT4RI and TFQI-FT4 showed positive associations with iSAP risk, while TFQI-FT3 retained overall significance in the fully adjusted model.

## Predictive performance of FT3/FT4 ratio and A$_2$DS$_2$ score

Receiver operating characteristic (ROC) analysis was performed to evaluate the predictive accuracy of the FT3/FT4 ratio (Model 1) and the A$_2$DS$_2$ score (Model 2) for iSAP. Model 1 demonstrated an area under the curve (AUC) of 0.7112 (95% CI: 0.6801–0.7423), with an optimal cut-off value of −1.2474, yielding a sensitivity of 72.07% and specificity of 61.39%. In comparison, Model 2 showed a slightly higher AUC of 0.7632 (95% CI: 0.7344–0.7920) with a cut-off value of −1.3855, achieving a sensitivity of 65.96% and specificity of 73.26% (Fig 2). The difference in AUCs between the two models was statistically significant (p = 0.0083).

Net reclassification improvement (NRI) and integrated discrimination improvement (IDI) analyses suggested limited reclassification benefits for Model 2 compared to Model 1. The NRI estimate was 0.0589 (p = 0.0892), while the IDI value was also 0.0589 (p = 0.0895). These findings indicate that while the A$_2$DS$_2$ score demonstrated higher overall predictive accuracy, the FT3/FT4 ratio may offer complementary value, particularly in scenarios requiring high sensitivity.

## Discussion

This study explores the association between thyroid hormone sensitivity, especially the FT3/FT4 ratio, and iSAP risk in euthyroid ischemic stroke patients. Our findings indicate a statistical association between a lower FT3/FT4 ratio and increased odds of iSAP, even after adjusting for demographic factors (e.g., age, sex), vascular comorbidities (e.g., smoking, hypertension, diabetes mellitus, atrial fibrillation, COPD), and key laboratory parameters (e.g., FPG, HDL-C, LDL-C, CRP, BUN, UA, HCY, WBC, eGFR, HbA1c, TG, AST) in multivariate models. Patients with iSAP exhibited

**Table 3. Multivariate Logistic Regression Analysis of Thyroid Hormone Sensitivity Indices and iSAP Risk.**

| Exposure | Crude Model (Model 1) | | Partially Adjusted Model (Model 2) | | Fully Adjusted Model (Model 3) | |
|---|---|---|---|---|---|---|
| | OR (95% CI) | P-value | OR (95% CI) | P-value | OR (95% CI) | P-value |
| Central TH sensitivity | | | | | | |
| TSHI | 0.84 (0.75, 0.95) | 0.0042 | 0.81 (0.72, 0.92) | 0.0009 | 0.92 (0.79, 1.08) | 0.3079 |
| Q1 (<1.57) | 1.0 | | 1.0 | | 1.0 | |
| Q2 (1.57~2.21) | 0.84 (0.62, 1.15) | 0.2770 | 0.82 (0.60, 1.13) | 0.2332 | 0.81 (0.53, 1.22) | 0.3100 |
| Q3 (2.21~2.75) | 0.63 (0.46, 0.87) | 0.0056 | 0.60 (0.43, 0.85) | 0.0032 | 0.80 (0.52, 1.22) | 0.2922 |
| Q4 (>2.75) | 0.73 (0.53, 1.01) | 0.0542 | 0.71 (0.51, 0.98) | 0.0368 | 0.95 (0.63, 1.43) | 0.8013 |
| TT4RI | 1.00 (0.99, 1.00) | 0.2514 | 1.00 (0.99, 1.00) | 0.1413 | 1.00 (1.00, 1.01) | 0.7094 |
| Q1 (<10.99) | 1.0 | | 1.0 | | 1.0 | |
| Q2 (10.99~20.26) | 0.85 (0.62, 1.15) | 0.2834 | 0.80 (0.58, 1.10) | 0.1707 | 0.82 (0.54, 1.23) | 0.3306 |
| Q3 (20.28~33.11) | 0.53 (0.38, 0.74) | 0.0002 | 0.51 (0.36, 0.71) | <0.0001 | 0.72 (0.47, 1.11) | 0.1375 |
| Q4 (>33.17) | 0.70 (0.51, 0.96) | 0.0258 | 0.66 (0.48, 0.92) | 0.0132 | 0.95 (0.63, 1.44) | 0.8247 |
| TFQI-FT3 | 0.08 (0.04, 0.15) | <0.0001 | 0.09 (0.05, 0.17) | <0.0001 | 0.35 (0.18, 0.67) | 0.0017 |
| Q1 (<−0.25) | 1.0 | | 1.0 | | 1.0 | |
| Q2 (−0.25~−0.15) | 0.34 (0.25, 0.47) | <0.0001 | 0.36 (0.26, 0.49) | <0.0001 | 0.59 (0.40, 0.88) | 0.0099 |
| Q3 (−0.15~0.01) | 0.26 (0.19, 0.36) | <0.0001 | 0.29 (0.21, 0.41) | <0.0001 | 0.67 (0.45, 1.02) | 0.0594 |
| Q4 (>0.01) | 0.22 (0.16, 0.31) | <0.0001 | 0.23 (0.16, 0.33) | <0.0001 | 0.44 (0.29, 0.68) | 0.0002 |
| TFQI-FT4 | 1.74 (1.08, 2.78) | 0.0216 | 1.67 (1.03, 2.72) | 0.0390 | 1.57 (0.85, 2.89) | 0.1512 |
| Q1 (<−0.16) | 1.0 | | 1.0 | | 1.0 | |
| Q2 (−0.16~0.01) | 1.18 (0.84, 1.65) | 0.3387 | 1.12 (0.79, 1.58) | 0.5248 | 1.17 (0.76, 1.80) | 0.4797 |
| Q3(−0.01~0.14) | 1.47 (1.06, 2.03) | 0.0213 | 1.34 (0.96, 1.88) | 0.0841 | 1.42 (0.93, 2.16) | 0.1019 |
| Q4 (>0.14) | 1.34 (0.97, 1.87) | 0.0794 | 1.28 (0.91, 1.81) | 0.1498 | 1.30 (0.85, 2.00) | 0.2253 |
| Peripheral TH sensitivity | | | | | | |
| FT3/FT4 | 0.80 (0.69, 0.92) | 0.0018 | 0.80 (0.69, 0.92) | 0.0016 | 0.87 (0.75, 1.01) | 0.0758 |
| Q1 (<0.25) | 1.0 | | 1.0 | | 1.0 | |
| Q2 (0.25~0.30) | 0.42 (0.32, 0.57) | <0.0001 | 0.45 (0.33, 0.61) | <0.0001 | 0.70 (0.48, 1.03) | 0.0692 |
| Q3 (0.30~0.37) | 0.18 (0.13, 0.25) | <0.0001 | 0.20 (0.14, 0.29) | <0.0001 | 0.40 (0.26, 0.62) | <0.0001 |
| Q4 (>0.37) | 0.16 (0.11, 0.23) | <0.0001 | 0.17 (0.12, 0.25) | <0.0001 | 0.31 (0.19, 0.48) | <0.0001 |

This table presents the results of multivariate logistic regression analyses evaluating the association between thyroid hormone sensitivity indices and iSAP risk. Three models were constructed: Model 1 is the crude model without adjustment; Model 2 is adjusted for age and sex; Model 3 is fully adjusted for age, sex, smoking status, hypertension, diabetes mellitus, atrial fibrillation (AF), chronic obstructive pulmonary disease (COPD), fasting plasma glucose (FPG), high-density lipoprotein cholesterol (HDL-C), low-density lipoprotein cholesterol (LDL-C), C-reactive protein (CRP), blood urea nitrogen (BUN), uric acid (UA), homocysteine (HCY), white blood cell count (WBC), estimated glomerular filtration rate (eGFR), glycated hemoglobin A1c (HbA1c), triglycerides (TG), and aspartate aminotransferase (AST).

Data are expressed as odds ratios (ORs) with 95% confidence intervals (CIs) and corresponding p-values.

Abbreviations are as defined in Table 1.

lower FT3 levels, elevated FT4 concentrations, and a reduced FT3/FT4 ratio, suggesting impaired peripheral thyroid hormone conversion. Furthermore, receiver operating characteristic (ROC) curve analysis indicated that the FT3/FT4 ratio had moderate predictive value for iSAP, with an area under the curve (AUC) of 0.7112 (95% CI: 0.6801–0.7423), a sensitivity of 72.07%, and a specificity of 61.39%. Although the FT3/FT4 ratio showed a slightly lower AUC than the $A_2DS_2$ score (0.7112 vs. 0.7632), it reflects distinct endocrine-immune interactions. This suggests that the FT3/FT4 ratio may serve as a complementary biomarker rather than a replacement. Future studies are warranted to assess whether integrating the FT3/FT4 ratio with established clinical scores like $A_2DS_2$ could enhance risk stratification

**Table 4. Multivariate Logistic Regression Analysis of Associations Between Thyroid Hormone Sensitivity Indices and iSAP Risk in the Matched Cohort.**

| Exposure | Crude Model (Model 1) | | Partially Adjusted Model (Model 2) | | Fully Adjusted Model (Model 3) | |
|---|---|---|---|---|---|---|
| | OR (95% CI) | *P*-value | OR (95% CI) | *P*-value | OR (95% CI) | *P*-value |
| Central TH sensitivity | | | | | | |
| TSHI | 0.97 (0.90, 1.04) | 0.3564 | 0.95 (0.88, 1.02) | 0.1425 | 0.98 (0.90, 1.05) | 0.5119 |
| Q1 | 1.0 | | 1.0 | | 1.0 | |
| Q2 | 1.44 (1.17, 1.77) | 0.0006 | 1.46 (1.18, 1.80) | 0.0004 | 1.55 (1.24, 1.93) | 0.0001 |
| Q3 | 0.88 (0.71, 1.07) | 0.2050 | 0.88 (0.72, 1.08) | 0.2169 | 0.96 (0.77, 1.19) | 0.6863 |
| Q4 | 1.12 (0.91, 1.37) | 0.2958 | 1.07 (0.87, 1.32) | 0.4959 | 1.15 (0.93, 1.43) | 0.2005 |
| TT4RI | 1.00 (1.00, 1.00) | 0.0348 | 1.00 (1.00, 1.00) | 0.1170 | 1.00 (1.00, 1.01) | 0.0098 |
| Q1 | 1.0 | | 1.0 | | 1.0 | |
| Q2 | 1.63 (1.32, 2.00) | <0.0001 | 1.64 (1.33, 2.02) | <0.0001 | 1.74 (1.39, 2.17) | <0.0001 |
| Q3 | 0.81 (0.66, 0.99) | 0.0433 | 0.81 (0.66, 1.00) | 0.0524 | 0.89 (0.71, 1.10) | 0.2746 |
| Q4 | 1.26 (1.03, 1.55) | 0.0264 | 1.21 (0.98, 1.49) | 0.0717 | 1.32 (1.06, 1.64) | 0.0138 |
| TFQI-FT3 | 1.34 (1.05, 1.72) | 0.0193 | 1.45 (1.12, 1.86) | 0.0041 | 1.51 (1.15, 1.99) | 0.0031 |
| Q1 | 1.0 | | 1.0 | | 1.0 | |
| Q2 | 0.75 (0.61, 0.92) | 0.0056 | 0.79 (0.64, 0.97) | 0.0272 | 0.87 (0.70, 1.08) | 0.2003 |
| Q3 | 0.57 (0.47, 0.71) | <0.0001 | 0.60 (0.49, 0.75) | <0.0001 | 0.64 (0.51, 0.80) | 0.0001 |
| Q4 | 1.11 (0.90, 1.36) | 0.3356 | 1.16 (0.94, 1.43) | 0.1628 | 1.23 (0.98, 1.53) | 0.0742 |
| TFQI-FT4 | 0.81 (0.61, 1.08) | 0.1528 | 0.74 (0.55, 0.99) | 0.0422 | 0.83 (0.61, 1.13) | 0.2444 |
| Q1 | 1.0 | | 1.0 | | 1.0 | |
| Q2 | 1.00 (0.81, 1.22) | 0.9790 | 0.96 (0.78, 1.19) | 0.7337 | 1.01 (0.81, 1.26) | 0.9251 |
| Q3 | 1.18 (0.96, 1.45) | 0.1129 | 1.12 (0.91, 1.38) | 0.2763 | 1.27 (1.02, 1.57) | 0.0344 |
| Q4 | 0.91 (0.74, 1.12) | 0.3667 | 0.85 (0.69, 1.05) | 0.1275 | 0.92 (0.74, 1.15) | 0.4786 |
| Peripheral TH sensitivity | | | | | | |
| FT3.FT4 | 1.07 (1.00, 1.14) | 0.0452 | 1.08 (1.02, 1.16) | 0.0153 | 1.06 (0.99, 1.14) | 0.0759 |
| Q1 | 1.0 | | 1.0 | | 1.0 | |
| Q2 | 0.61 (0.50, 0.75) | <0.0001 | 0.64 (0.52, 0.79) | <0.0001 | 0.69 (0.56, 0.86) | 0.0011 |
| Q3 | 0.78 (0.64, 0.96) | 0.0194 | 0.84 (0.68, 1.04) | 0.1136 | 0.86 (0.69, 1.09) | 0.2127 |
| Q4 | 0.67 (0.54, 0.82) | 0.0001 | 0.73 (0.59, 0.90) | 0.0039 | 0.72 (0.57, 0.91) | 0.0054 |

This table presents the results of multivariate logistic regression models examining the association between various thyroid hormone sensitivity indices and the risk of ischemic stroke-associated pneumonia (iSAP) in the propensity score–matched cohort (n = 2926). Odds ratios (ORs) and 95% confidence intervals (CIs) are shown for both continuous models and quartile-based categorical models. Three models were constructed: Model 1 was unadjusted (crude), Model 2 was adjusted for age and sex, and Model 3 was fully adjusted for demographic and clinical covariates. Statistically significant associations (p < 0.05) are highlighted in bold.

Three models were constructed: Model 1 is the crude model without adjustment; Model 2 is adjusted for age and sex; Model 3 is fully adjusted for age, sex, smoking status, hypertension, diabetes mellitus, atrial fibrillation (AF), chronic obstructive pulmonary disease (COPD), fasting plasma glucose (FPG), high-density lipoprotein cholesterol (HDL-C), low-density lipoprotein cholesterol (LDL-C), C-reactive protein (CRP), blood urea nitrogen (BUN), uric acid (UA), homocysteine (HCY), white blood cell count (WBC), estimated glomerular filtration rate (eGFR), glycated hemoglobin A1c (HbA1c), triglycerides (TG), and aspartate aminotransferase (AST).

Data are expressed as odds ratios (ORs) with 95% confidence intervals (CIs) and corresponding p-values.

Abbreviations are as defined in Table 1.

models for iSAP. These findings may support further investigation of the FT3/FT4 ratio as a potential adjunct marker in early iSAP risk assessment, particularly in patients who may not be readily identified using conventional risk stratification tools.

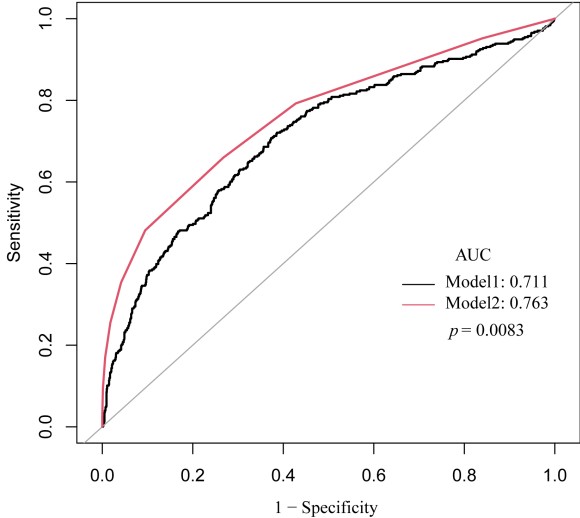

**Fig 2. Receiver operating characteristic (ROC) curve comparison of FT3/FT4 ratio and A₂DS₂ score for iSAP prediction.** This figure presents the ROC analysis evaluating the predictive performance of the FT3/FT4 ratio (Model 1) and the A₂DS₂ score (Model 2) for ischemic stroke-associated pneumonia (iSAP). Model 1 (black curve) demonstrated an area under the curve (AUC) of 0.7112 (95% CI: 0.6801–0.7423), with an optimal cut-off value of −1.2474, yielding a sensitivity of 72.07% and specificity of 61.39%. Model 2 (red curve) exhibited a slightly higher AUC of 0.7632 (95% CI: 0.7344–0.7920) with a cut-off value of −1.3855, achieving a sensitivity of 65.96% and specificity of 73.26%. The difference between the two AUCs was statistically significant (p = 0.0083).

Our study identified a significant association between a lower FT3/FT4 ratio and an increased risk of ischemic stroke-associated pneumonia (iSAP), which may be explained by the multifaceted role of thyroid hormones in immune regulation, neuromuscular function, and metabolic adaptation. Thyroid hormones, particularly FT3, are critical for maintaining immune homeostasis by modulating T-cell proliferation, macrophage phagocytosis, and inflammatory cytokine release [14]. A low FT3 state has been linked to impaired immune cell function, resulting in increased production of pro-inflammatory cytokines such as TNF-α and IL-6, which can exacerbate systemic inflammation, compromise pulmonary defense barriers, and heighten susceptibility to infection [15]. Additionally, reduced FT3 levels may impair neutrophil chemotaxis and macrophage phagocytic activity [16], further weakening host defenses and increasing the likelihood of iSAP.

Beyond immune function, dysphagia is a major contributor to iSAP in stroke patients [17–19], and a lower FT3/FT4 ratio may exacerbate neuromuscular dysfunction, further elevating aspiration risk [20]. Thyroid hormones play a pivotal role in the maintenance of swallowing and respiratory muscle function [21], and a low FT3 state has been associated with delayed swallowing reflexes and reduced pharyngeal muscle contractility, increasing the probability of aspiration [22]. Furthermore, previous studies [23] link low T3 levels with impaired neutrophil function, reduced cytokine responses, and weakened pulmonary defenses, potentially increasing infection risk. Although serum TSH levels differed slightly between the iSAP and non-iSAP groups at baseline, this difference did not translate into an independent association with iSAP risk in multivariate analyses. This finding aligns with the concept of non-thyroidal illness syndrome (NTIS), wherein peripheral thyroid hormone levels, particularly FT3 and FT4, exhibit more dynamic fluctuations during acute illness, while TSH levels remain relatively stable [24]. Thus, peripheral hormone indices may serve as more sensitive indicators of iSAP risk compared to TSH in the acute phase of ischemic stroke. These findings raise the hypothesis that alterations in thyroid hormone sensitivity may play a role in iSAP pathophysiology through immune dysfunction, neuromuscular impairment, and impaired pulmonary defense mechanisms [25–28]. Previous studies have further demonstrated that low T3 levels impair neutrophil function, reduce cytokine responses, and compromise pulmonary host defenses, which may increase susceptibility to infections [29,30], providing mechanistic support for our observations.

In addition to the biological plausibility, it was important to account for potential clinical confounders in our analysis. Moreover, key comorbidities known to influence both thyroid function and iSAP risk, including atrial fibrillation and chronic obstructive pulmonary disease (COPD), were incorporated as covariates in the multivariable logistic regression models to minimize confounding effects.

Another potential mechanism involves non-thyroidal illness syndrome (NTIS), also referred to as low T3 syndrome, a condition frequently observed in acute stroke patients that may reflect a physiological stress response [24,29]. Stroke-induced inflammation may suppress deiodinase activity [30–32], impairing T4-to-T3 conversion and lowering the FT3/FT4 ratio [32]. Although initially adaptive, prolonged low T3 impairs protein synthesis, tissue repair, and immunity, increasing infection risk [33].

Additionally, low T3 levels have been linked to elevated oxidative stress, which may further compromise the pulmonary epithelial barrier, making the lungs more susceptible to bacterial invasion [34]. The interplay between metabolic dysregulation, immune dysfunction, and neuromuscular impairment suggests that a lower FT3/FT4 ratio may serve as a biomarker of physiological stress and a contributing factor in iSAP pathogenesis [35–37]. Given these mechanisms, the FT3/FT4 ratio may represent a potential risk indicator and hypothesis-generating target, warranting further investigation in prospective interventional studies [38,39].

Our study evaluated the predictive performance of the FT3/FT4 ratio and the $A_2DS_2$ score for ischemic stroke-associated pneumonia (iSAP) using ROC curve analysis, revealing distinct strengths and limitations for each model. The results suggest that the FT3/FT4 ratio could help identify high-risk patients often missed by conventional scoring systems. A comparative study [40] showed that ROC-based models effectively distinguished high- and low-risk iSAP patients, emphasizing the role of immune-inflammatory markers in enhancing predictive models. Similarly, a study [41]developing a predictive nomogram for iSAP in post-thrombectomy patients reported an AUC exceeding 0.80, underscoring the potential of integrating laboratory biomarkers-such as inflammatory markers and thyroid hormone indices-to enhance traditional clinical risk scores.

Beyond conventional scoring systems like the $A_2DS_2$ score, machine learning-based models have increasingly been applied to iSAP risk prediction [41–43], with some achieving AUC values between 0.84 and 0.90, surpassing traditional logistic regression models. These models leverage high-dimensional data, incorporating variables such as systemic inflammation, dysphagia severity, and metabolic dysfunction, further supporting the potential value of thyroid hormone indices as novel predictors of post-stroke pneumonia. Given the complementary strengths of these models, a hybrid approach incorporating thyroid hormone sensitivity indices into established scoring systems may significantly improve iSAP risk prediction.

This study has several notable strengths. It is among the first to systematically examine the relationship between thyroid hormone sensitivity and iSAP risk, contributing to a broader understanding of endocrine-immune interactions in stroke complications. Additionally, the large sample size and adjusted analyses strengthen our findings by minimizing potential confounders. However, several limitations should be acknowledged. First, the retrospective nature of the study limits the ability to establish causal relationships. While we observed significant associations between thyroid hormone sensitivity indices and iSAP risk, correlation does not imply causation. Second, the data were derived from a single-center cohort, which may limit the generalizability of our findings to other populations. Given that this study was conducted in a single hospital in China, future multicenter studies involving more diverse patient populations are needed to validate the applicability of these results across different contexts. Third, the study population was limited to euthyroid patients, and individuals with thyroid dysfunction were excluded. As a result, our findings may not be generalizable to patients with underlying thyroid disease. Additionally, thyroid hormone levels were measured only once upon admission, preventing the assessment of dynamic post-stroke changes, which might influence the evolution of thyroid function over time. Furthermore, although COPD was included as a covariate in our multivariate model, it was treated as a binary variable based on documented diagnosis and lacked granularity regarding disease severity. This limitation may have led to residual confounding

related to the systemic inflammatory burden associated with more advanced COPD. Future prospective studies incorporating serial thyroid function assessments are needed to validate our findings. Although age was statistically associated with iSAP risk (OR = 1.05), the effect size was modest and may not reflect strong clinical significance on an individual level. Nonetheless, age remains a well-established population-level risk factor and was appropriately adjusted for in our multivariate model.

Our results underscore the need for further investigation. Prospective multicenter studies are needed to validate the FT3/FT4 ratio's predictive value for iSAP risk across diverse populations. Furthermore, randomized controlled trials (RCTs) investigating T3 supplementation or metabolic interventions could clarify whether these strategies reduce iSAP risk and improve stroke outcomes. These studies are vital to confirming the broader applicability and clinical utility of our findings. Integrating thyroid hormone indices, such as the FT3/FT4 ratio, into established clinical risk models like the $A_2DS_2$ score could improve iSAP risk stratification. These indices may serve as additional biomarkers to refine the assessment of stroke patients at risk for pneumonia. Future studies, including prospective multicenter trials, are needed to explore this integration and evaluate its impact on clinical decision-making and patient outcomes. However, given the retrospective design, the observed associations between thyroid hormone sensitivity indices and iSAP should be interpreted as hypothesis-generating rather than implying causality. In particular, the inverse relationship between FT3 levels and iSAP risk may reflect a marker of disease severity rather than a causal factor. Therefore, longitudinal cohort studies are warranted to verify the temporal and mechanistic links.

## Conclusions

In conclusion, this study generates new hypotheses regarding the role of thyroid hormone sensitivity indices, particularly the FT3/FT4 ratio, in iSAP risk stratification. Given the observed inverse association between FT3/FT4 ratios and iSAP risk, future studies should evaluate whether incorporating thyroid markers into established risk models can enhance predictive accuracy and validate these findings in longitudinal cohorts.

## Supporting information

**S1 Table. Comparison of Standardized Mean Differences (SMDs) of Covariates Before and After Propensity Score Matching (PSM) Combined with Genetic Matching (GenMatch).** Supplementary Table 1 presents the changes in standardized mean differences (SMDs) for baseline covariates before and after 1:1 matching between the iSAP (n = 376) and non-iSAP (n = 1391) groups using a combination of propensity score matching (PSM) and Genetic Matching (GenMatch) algorithms. After matching, both groups comprised 1463 patients each. Most covariates showed a marked reduction in SMDs, with values falling below 0.1, indicating satisfactory baseline balance. Abbreviations are as defined in Table 1. Additional abbreviations: SMD, Standardized Mean Difference; PSM, Propensity Score Matching; GenMatch, Genetic Matching.
(DOCX)

**S1 Fig. Standardized Mean Differences of Baseline Covariates Before and After Propensity Score Matching with Genetic Matching Algorithm.** This figure displays the standardized mean differences (SMDs) for baseline covariates between the ischemic stroke-associated pneumonia (iSAP) group and the non-iSAP group before and after propensity score matching (PSM) using the Genetic Matching algorithm. The horizontal axis represents the absolute value of SMDs, with a commonly accepted threshold of 0.1 (dashed vertical line) indicating adequate balance. Covariate balance substantially improved after matching, as evidenced by most post-matching SMDs falling below the 0.1 threshold. This demonstrates that the matched sample achieved a high degree of comparability between groups, thereby minimizing potential confounding bias in subsequent analyses.
(DOCX)

**S1 Abbreviations.** List of Abbreviations.

(DOCX)

## Author contributions

**Conceptualization:** Zhidan Hua, Zichen Rao, Yiming Zhang, Chunyan Zhu.

**Data curation:** Yiming Zhang, Chunyan Zhu.

**Formal analysis:** Zichen Rao.

**Investigation:** Yiming Zhang, Chunyan Zhu.

**Methodology:** Zhidan Hua.

**Project administration:** Zhidan Hua, Zichen Rao.

**Resources:** Zhidan Hua.

**Software:** Zichen Rao.

**Supervision:** Zhidan Hua, Yiming Zhang, Chunyan Zhu.

**Validation:** Zhidan Hua, Zichen Rao.

**Visualization:** Zichen Rao, Yiming Zhang, Chunyan Zhu.

**Writing – original draft:** Zichen Rao.

**Writing – review & editing:** Zhidan Hua, Zichen Rao, Yiming Zhang, Chunyan Zhu.

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
