## [Decision Letter · Decision Letter 0]

24 Apr 2025

PONE-D-25-07372Association Between Thyroid Hormone Sensitivity and Ischemic Stroke-Associated Pneumonia: The Role of FT3/FT4 RatioPLOS ONE

Dear Dr. Rao,

Thank you for submitting your manuscript to PLOS ONE. After careful consideration, we feel that it has merit but does not fully meet PLOS ONE’s publication criteria as it currently stands. Therefore, we invite you to submit a revised version of the manuscript that addresses the points raised during the review process.

We look forward to receiving your revised manuscript.

Kind regards,

Ennio Polilli

Academic Editor

PLOS ONE

Reviewers' comments:

Reviewer's Responses to Questions

**Comments to the Author**

1. Is the manuscript technically sound, and do the data support the conclusions?

Reviewer #1: Partly

Reviewer #2: Partly

Reviewer #3: Partly

2. Has the statistical analysis been performed appropriately and rigorously? 

Reviewer #1: I Don't Know

Reviewer #2: No

Reviewer #3: Yes

3. Have the authors made all data underlying the findings in their manuscript fully available?

Reviewer #1: Yes

Reviewer #2: No

Reviewer #3: Yes

4. Is the manuscript presented in an intelligible fashion and written in standard English?

Reviewer #1: Yes

Reviewer #2: Yes

Reviewer #3: No

5. Review Comments to the Author

Reviewer #1: General comments:

The manuscript PONE-D-25-07372 is interesting and raises a good scientific question. The correlation of FT3/FT4 ratio with other illnesses, namely cardiovascular disease, metabolic disease, and diabetes has been previously shown. The association between thyroid hormone regulation and iSAP risk has been recently addressed (doi: 10.3389/fendo.2024.1438700.). The manuscript includes a large patient cohort, which enhances statistical analysis and improves the reliability of the results. A key strength of this manuscript is its use of multivariate adjusted analysis in a large cohort.

My major concern with the manuscript is that there is no description of how the statistical analysis was performed, or which statistical tests were conducted. Furthermore, the result interpretation in the univariate analysis of FT3 and FT3/FT4 ratios is confusing and unclear. As the authors acknowledge, another limitation of this study is the single-time measurement of hormone levels. A follow-up assay incorporating multiple timepoints should be conducted in the future to strengthen the findings. Additionally, the authors should carefully review the manuscript for minor writing errors, particularly missing spaces between words and punctuation (e.g., in Definitions, line 14: “auscultation.Impaired”), among others. Please ensure thorough proofreading.

Overall, I believe this is a strong manuscript and recommend publication after revisions.

Materials and Methods:

Study design and participants. How were the control patients selected? Were the cases and controls paired? Please clarify in this section.

Statistical analysis: Please provide details on how the statistical analysis was conducted. What were the applied statistical tests? Where these parametric or non-parametric tests? Regarding multivariate analysis, which confounding variables were considered in the models? How were these obtained?

Results:

Tables 1: the number of patients in each group (case/control) for each analysis characteristic should be mentioned. This is very important information that supports the statistical analysis. Please complete the table.

Baseline characteristics: Please cite Table 1 in paragraph 2.

Thyroid hormone profiles: Please cite Table 1 in both paragraphs.

Association between thyroid hormone sensitivity and iSAP risk: The OR of 1.05 for the age difference is statistically significant, however, in clinical terms does not seem relevant.

The section “Regarding thyroid hormone parameters, lower FT3 levels (OR = 0.85, 95% CI: 0.80-0.91, p < 0.0001) and a reduced FT3/FT4 ratio (OR = 0.80, 95% CI: 0.69-0.92, p = 0.0018) were also significantly associated with increased iSAP risk.” Is my main concern. I am having some trouble interpreting the OR values. Usually, lower OR represents lower risk, rather than higher risk. Please clarify how this analysis was made and confirm that no errors were made.

Table 3: Please add more information on how the analysis of Table 3 was conducted. Clarify Q1, Q2, Q3 and Q4. Indicate the included variables.

Discussion:

Please clarify what you mean in the first paragraph, line 5, by “after adjusting for multiple confounders”.

Supplementary data:

Please place the abbreviations by alphabetical order.

Reviewer #2: 1- the statistical analysis not mentioned ? 2- what are the reference ranges and units of the analytes studied? How was the normal range of thyroid ratio sensitivity determined ? 4- can you compare results with hypothyroidism and hyperthyroidism? 5- can you discuss the role of TSH in This study in more detail?

Reviewer #3: Strengths:

1. Novelty: The study addresses an underexplored topic by investigating the association between thyroid hormone sensitivity, particularly the FT3/FT4 ratio, and ischemic stroke-associated pneumonia (iSAP) in euthyroid patients.

2. Methodological Rigor: The study includes a large sample size (1,767 euthyroid stroke patients) and employs both univariate and multivariate logistic regression to adjust for confounding factors.

3. Statistical Analysis: ROC curve analysis enhances the study by demonstrating the predictive value of the FT3/FT4 ratio compared to the A2DS2 score.

4. Clinical Relevance: The identification of the FT3/FT4 ratio as a potential biomarker could aid in the early identification of patients at risk for iSAP.

Major Concerns:

1. Study Design and Data Collection:

o The retrospective nature of the study inherently limits causality. The authors should discuss this limitation more thoroughly, emphasizing that correlation does not imply causation.

o Thyroid hormone levels were measured only once, which may not capture dynamic changes in thyroid function following stroke. Including this limitation in the discussion would improve the manuscript.

2. Statistical Validity:

o The difference in AUC between the FT3/FT4 ratio and the A2DS2 score is relatively small (0.711 vs. 0.763). The clinical significance of this difference should be addressed more clearly, as the practical advantage of using the FT3/FT4 ratio over established methods remains uncertain.

3. Interpretation of Results:

o The discussion on thyroid hormone sensitivity in the pathophysiology of iSAP is speculative. Providing more references or mechanistic insights to support these hypotheses would strengthen the conclusions.

o It remains unclear how comorbidities like atrial fibrillation and COPD, which affect both thyroid function and iSAP risk, were factored into the analysis. Clarifying this aspect would enhance the manuscript's validity.

4. Generalizability:

o The study is conducted at a single center in China, which may limit the applicability of findings to other populations. Explicitly discussing this limitation would improve transparency.

Minor Concerns:

1. Language and Clarity:

o Some sections contain long and complex sentences that could be simplified for better readability. A thorough proofreading to correct minor grammatical errors and improve sentence structure would be beneficial.

2. Data Presentation:

o The tables and figures are adequate but could include more interpretation within the captions to enhance understanding.

o Including a flowchart summarizing the patient inclusion and exclusion process would improve the manuscript's reproducibility and clarity.

3. Future Directions:

o The manuscript lacks a clear outline of future research. Suggestions for prospective multicenter cohort studies or interventional trials investigating thyroid modulation would add value.

o Including a discussion on integrating thyroid hormone indices into established clinical risk models would be beneficial.

6. PLOS authors have the option to publish the peer review history of their article (what does this mean? ). If published, this will include your full peer review and any attached files.

**Do you want your identity to be public for this peer review?** For information about this choice, including consent withdrawal, please see our Privacy Policy .

Reviewer #1: No

Reviewer #2: No

Reviewer #3: No

---

## [Author Response · Author response to Decision Letter 1]

29 Apr 2025

Title: Response to Reviewers

Manuscript ID: PONE-D-25-07372

Title: Association Between Thyroid Hormone Sensitivity and Ischemic Stroke-Associated Pneumonia: The Role of FT3/FT4 Ratio

General Response to the Editor and Reviewers

We sincerely thank the Editors and all Reviewers for their careful evaluation and constructive feedback on our manuscript. We have carefully addressed each comment and made corresponding revisions to improve the clarity, scientific rigor, and overall quality of the paper. Below, we provide point-by-point responses to all comments, with detailed explanations of the revisions made.

Response to Reviewer #1, Comment: My major concern with the manuscript is that there is no description of how the statistical analysis was performed, or which statistical tests were conducted.

Response: We appreciate your attention to the statistical rigor of our manuscript, which enabled us to enhance the precision and detail of this section. We are grateful for your critical evaluation and agree that a more detailed description of our statistical approach is warranted. In the revised manuscript, we have included a comprehensive description of the statistical methods employed, specifically in the “Statistical Analysis” subsection of the Materials and Methods section.

In brief, continuous variables were tested for normality using the Kolmogorov–Smirnov test. Depending on the distribution, either Student’s t-tests or Mann–Whitney U tests were used for group comparisons. Categorical variables were compared using the Chi-square test or Fisher’s exact test where appropriate. Univariate logistic regression analysis was initially conducted to identify potential risk factors for iSAP. Subsequently, three multivariate logistic regression models were constructed: Model 1 was unadjusted (crude); Model 2 was partially adjusted for age and NIHSS score; and Model 3 was fully adjusted for multiple clinical covariates, including demographic factors (age, sex), vascular risk factors (smoking, hypertension, diabetes, atrial fibrillation [AF], chronic obstructive pulmonary disease [COPD]), neurological status (GCS score, KWDT score), and laboratory parameters (FPG, HDL-C, LDL-C, CRP, BUN, UA, HCY, WBC, eGFR, HbA1c, TG, AST). We have also added clarification regarding the interpretation of OR values for FT3 and FT3/FT4 in the results and discussion sections. Specifically, since lower FT3/FT4 values indicate reduced peripheral conversion efficiency, an OR < 1 reflects a higher risk of iSAP associated with decreased thyroid hormone sensitivity.

The inserted paragraph

Statistical analyses were conducted using R Studio (version 4.2.2; R Foundation for Statistical Computing, Vienna, Austria) and EmpowerStats (version 2.0; www.empowerstats.com). Continuous variables were assessed for normality using the Kolmogorov–Smirnov test. Normally distributed variables were expressed as means ± standard deviation (SD) and compared using independent Student’s t-tests. Non-normally distributed variables were presented as medians with interquartile ranges (IQRs) and compared using the Mann-Whitney U test. Categorical variables were expressed as counts and percentages and compared using Chi-square or Fisher’s exact test, as appropriate.

Univariate logistic regression analyses were first performed to screen for potential predictors of ischemic stroke-associated pneumonia (iSAP). Variables with a p-value < 0.1 in the univariate analysis were subsequently included in multivariate logistic regression models using a backward stepwise elimination approach. Three models were constructed: (Model 1) unadjusted (crude); (Model 2) adjusted for age and sex; and (Model 3) fully adjusted for demographic and clinical variables, including age, sex, smoking status, hypertension, diabetes mellitus, atrial fibrillation (AF), chronic obstructive pulmonary disease (COPD), fasting plasma glucose (FPG), high-density lipoprotein cholesterol (HDL-C), low-density lipoprotein cholesterol (LDL-C), C-reactive protein (CRP), blood urea nitrogen (BUN), uric acid (UA), homocysteine (HCY), white blood cell count (WBC), estimated glomerular filtration rate (eGFR), glycated hemoglobin A1c (HbA1c), triglycerides (TG), and aspartate aminotransferase (AST). Odds ratios (ORs) and 95% confidence intervals (CIs) were reported. A two-tailed p-value < 0.05 was considered statistically significant.

Receiver operating characteristic (ROC) curve analysis was conducted to evaluate the predictive performance of the FT3/FT4 ratio and the A2DS2 score in identifying patients at risk for iSAP. The area under the curve (AUC) was calculated for each model, and the DeLong test was used to compare differences between AUCs.

For details, see the revised version on pages 8-9, lines 170-195.

Response to Reviewer #1, Comment: As the authors acknowledge, another limitation of this study is the single-time measurement of hormone levels. A follow-up assay incorporating multiple timepoints should be conducted in the future to strengthen the findings.

Response: We sincerely thank you for emphasizing this important limitation. As noted in the revised Discussion section (Page 23, Lines 431-434), we acknowledged that the single-time measurement of thyroid hormone levels is a key limitation of our study. We also emphasized the need for future prospective studies incorporating serial thyroid function assessments to better elucidate dynamic changes over time and their potential impact on iSAP risk.

Response to Reviewer #1, Comment: Furthermore, the result interpretation in the univariate analysis of FT3 and FT3/FT4 ratios is confusing and unclear.

Response: Thank you for your thoughtful observation, which helped us refine the clarity of our analysis. We understand that the phrasing in the Results section regarding the interpretation of odds ratios (ORs) may have caused confusion. Specifically, although the OR values for FT3 (0.85) and FT3/FT4 ratio (0.80) are less than 1, our original wording may have implied the inverse interpretation. To clarify, we have revised the sentence to explicitly indicate that lower FT3 levels and lower FT3/FT4 ratios were associated with increased risk of iSAP, which is statistically reflected by the OR < 1. This has been clarified in both the Results and Discussion sections to enhance interpretability and avoid ambiguity.

Revised sentence (Results section):

“Regarding thyroid hormone parameters, lower FT3 levels were associated with increased odds of developing iSAP (OR = 0.85, 95% CI: 0.80–0.91, p < 0.0001), as was a lower FT3/FT4 ratio (OR = 0.80, 95% CI: 0.69–0.92, p = 0.0018). These findings suggest that each unit decrease in FT3 or FT3/FT4 ratio corresponds to a 15% and 20% increase in the odds of iSAP, respectively”

The specific changes are on page 13, lines 252-256.

Response to Reviewer #1, Comment: Additionally, the authors should carefully review the manuscript for minor writing errors, particularly missing spaces between words and punctuation (e.g., in Definitions, line 14: “auscultation.Impaired”), among others. Please ensure thorough proofreading.

Response: We greatly appreciate your careful reading and constructive feedback. We have now carefully proofread the entire text and corrected all identified issues, including the missing space in “auscultation. Impaired” (now corrected) and other similar formatting inconsistencies.

In addition, we performed a comprehensive language review to ensure accuracy, stylistic consistency, and appropriate spacing between words and punctuation. We appreciate your attention to detail, which helped improve the overall readability of our manuscript.

Response to Reviewer #1, Comment: Materials and Methods: Study design and participants. How were the control patients selected? Were the cases and controls paired? Please clarify in this section.

Response: We appreciate your thoughtful suggestion, which prompted a clearer revision. We agree that the original version of the manuscript did not clearly explain our study design and group assignment strategy. In response, we have revised the “Study Design and Participants” section to clarify that this study was based on a single-center retrospective cohort comprising 1,767 acute ischemic stroke (AIS) patients admitted between September 2016 and September 2022.

Specifically, we have added the following sentences on Page 3, Lines 82-87 of the revised manuscript: “All included patients were enrolled into a single-center retrospective cohort and were subsequently stratified into two groups based on whether they developed ischemic stroke-associated pneumonia (iSAP) during hospitalization: an iSAP group and a non-iSAP group. No case-control matching or pairing was performed; group differences were addressed through multivariate adjustment.”

This modification makes it clear that the study did not employ a case-control design or any form of matching. The iSAP and non-iSAP groups were derived post hoc from a single cohort, and potential confounding was handled through multivariate regression. We hope this revision fully addresses your concern and enhances the precision and completeness of the manuscript’s methodological reporting.

Response to Reviewer #1, Comment: Statistical Analysis: Statistical analysis: Please provide details on how the statistical analysis was conducted. What were the applied statistical tests? Were these parametric or non-parametric tests? Regarding multivariate analysis, which confounding variables were considered in the models? How were these obtained?

Response: Thank you for your meticulous follow-up comments. As detailed in our response to your earlier major comment and now clarified in the revised Statistical Analysis section (Page 8-9, Lines 170-195), we have added a comprehensive description of the statistical approach.

To specifically address your points: We clarified that parametric (Student’s t-test) and non-parametric (Mann-Whitney U test) methods were used for continuous variable comparisons, depending on normality (assessed via the Kolmogorov-Smirnov test).

We now explicitly list the confounders adjusted for in the fully adjusted multivariate model: age, sex, smoking status, hypertension, diabetes mellitus, atrial fibrillation (AF), chronic obstructive pulmonary disease (COPD), fasting plasma glucose (FPG), high-density lipoprotein cholesterol (HDL-C), low-density lipoprotein cholesterol (LDL-C), C-reactive protein (CRP), blood urea nitrogen (BUN), uric acid (UA), homocysteine (HCY), white blood cell count (WBC), estimated glomerular filtration rate (eGFR), glycated hemoglobin A1c (HbA1c), triglycerides (TG), and aspartate aminotransferase (AST).

We sincerely appreciate your detailed suggestions, which helped us refine the methodology and enhance the precision of our descriptions.

Response to Reviewer #1, Comment: Tables 1: the number of patients in each group (case/control) for each analysis characteristic should be mentioned. This is very important information that supports the statistical analysis. Please complete the table.”

Response: We highly value your careful review and helpful observation regarding the group-specific data presentation in Table 1. Recognizing the importance of this information for interpreting the statistical analyses, we have updated Table 1 (Page 11-13, Lines 226-246) in the revised manuscript to explicitly display the number of patients in each group (iSAP and non-iSAP) for every baseline characteristic.

This revision provides a clearer and more transparent overview of the cohort composition and supports the validity of the comparative analyses presented.

We thank you again for highlighting this important detail, which has enhanced the quality and precision of our Results section.

Response to Reviewer #1, Comment: Baseline characteristics: Please cite Table 1 in paragraph 2.

Response: We appreciate your valuable suggestion, which improved the structure of the Results section. In the revised manuscript, we have added an explicit citation to Table 1 when reporting the baseline characteristics of the iSAP and non-iSAP groups in the Results section (Page 10, Line 211). This allows readers to directly reference the corresponding data and improves the comprehensibility and traceability of the reported findings.

Response to Reviewer #1, Comment: Thyroid hormone profiles: Please cite Table 1 in both paragraphs.

Response: Thank you for highlighting this important point. To enhance the interpretability and traceability of our results, we have now inserted explicit references to Table 1 in both relevant paragraphs within the “Thyroid hormone profiles” section of the Results.

Specifically, these citations have been added on Page 11, Line 219 (after reporting FT3, FT4, FT3/FT4 ratio, and TSH levels) and Line 225 (in the paragraph discussing thyroid hormone sensitivity indices). These additions ensure that all reported hormone data can be directly cross-referenced with the values presented in Table 1.

Response to Reviewer #1, Comment: Association between thyroid hormone sensitivity and iSAP risk: The OR of 1.05 for the age difference is statistically significant, however, in clinical terms does not seem relevant.

Response: We truly value your thoughtful observation regarding the odds ratio for age. We agree that while this association was statistically significant, the magnitude of the effect is relatively modest and may have limited clinical relevance at the individual level.

To address this, we have added a clarifying statement in the Discussion section (Page 23, Lines 435-439), acknowledging the limited effect size and emphasizing the importance of interpreting this finding with clinical caution. We also note that age was included in the multivariate model as an established population-level risk factor. We believe this addition enhances the interpretability and balance of our findings.

Response to Reviewer #1, Comment: The section “Regarding thyroid hormone parameters, lower FT3 levels (OR = 0.85, 95% CI: 0.80-0.91, p < 0.0001) and a reduced FT3/FT4 ratio (OR = 0.80, 95% CI: 0.69-0.92, p = 0.0018) were also significantly associated with increased iSAP risk.” Is my main concern. I am having some trouble interpreting the OR values. Usually, lower OR represents lower risk, rather than higher risk. Please clarify how this analysis was made and confirm that no errors were made.

Response: Thank you for bringing this important point to our attention. We understand that the interpretation of odds ratios less than 1 may sometimes be counterintuitive. In our analysis, lower levels of FT3 and a reduced FT3/FT4 ratio were indeed associated with a higher risk of iSAP, which is reflected by OR values < 1.

To avoid confusion, we have revised the corresponding sentence in the Results section (Page 13, Lines 252-256) to clearly state that lower thyroid hormone levels were associated with increased odds of iSAP. We have also double-checked our statistical models to confirm that this directionality is correct. We hope this clarification resolves the concern.

Response to Reviewer #1, Comment: Table 3: Please add more information on how the analysis of Table 3 was conducted. Clarify Q1, Q2, Q3 and Q4. Indicate the included variables.

Response: We are grateful for your helpful and detailed comment. In response, we have made several clarifications to improve the openness and interpretive clarity of Table 3: We have now added the specific cutoff values for Q1–Q4 for all five thyroid-related parameters (THSI, TT4RI, TFQI-FT3, TFQI-FT4, and FT3/FT4 ratio) directly in the table. These ranges are based on quartile distributions within our study population.

Additionally, we have included a clear description in the table legend indicating that the analysis was conducted using multivariate logistic regression models (Model 3), with adjustments for age, sex, smoking status, hypertension, diabetes mellitus, atrial fibrillation (AF), chronic obstructive pulmonary disease (COPD), fasting plasma glucose (FPG), high-density lipoprotein cholesterol (HDL-C), low-density lipoprotein cholesterol (LDL-C), C-reactive protein (CRP), blood urea nitrogen (BUN), uric acid (UA), homocysteine (HCY), white blood cell count (WBC), estimated glomerular filtration rate (eGFR), g

---

## [Decision Letter · Decision Letter 1]

25 Jun 2025

PONE-D-25-07372R1Association Between Thyroid Hormone Sensitivity and Ischemic Stroke-Associated Pneumonia: The Role of FT3/FT4 RatioPLOS ONE

Dear Dr. Rao, 

Thank you for submitting your manuscript to PLOS ONE. After careful consideration, we feel that it has merit but does not fully meet PLOS ONE’s publication criteria as it currently stands. Therefore, we invite you to submit a revised version of the manuscript that addresses the points raised during the review process.

Please submit your revised manuscript by Aug 09 2025 11:59PM. If you will need more time than this to complete your revisions, please reply to this message or contact the journal office at plosone@plos.org . Please include the following items when submitting your revised manuscript:

We look forward to receiving your revised manuscript.

Kind regards,

Ennio Polilli

Academic Editor

PLOS ONE

Journal Requirements:

Reviewers' comments:

Reviewer's Responses to Questions

**Comments to the Author**

1. If the authors have adequately addressed your comments raised in a previous round of review and you feel that this manuscript is now acceptable for publication, you may indicate that here to bypass the “Comments to the Author” section, enter your conflict of interest statement in the “Confidential to Editor” section, and submit your "Accept" recommendation.

Reviewer #2: (No Response)

Reviewer #3: All comments have been addressed

Reviewer #4: (No Response)

2. Is the manuscript technically sound, and do the data support the conclusions?

Reviewer #2: Partly

Reviewer #3: Yes

Reviewer #4: Yes

3. Has the statistical analysis been performed appropriately and rigorously? 

Reviewer #2: No

Reviewer #3: Yes

Reviewer #4: Yes

4. Have the authors made all data underlying the findings in their manuscript fully available?

Reviewer #2: No

Reviewer #3: Yes

Reviewer #4: No

5. Is the manuscript presented in an intelligible fashion and written in standard English?

Reviewer #2: Yes

Reviewer #3: Yes

Reviewer #4: No

6. Review Comments to the Author

Reviewer #2: The paper discusses an important point regarding thyroid hormones in stroke pateints, but the methods used to address the problem need to be improved, for example the difference between the control and study group number is large ( iSAP group (n = 376) and the control group (n = 1,391). The eGFR values need to be repeated, what are the normal ranges of eGFR? Did you use serum or plasma for biomarker measurements? The ratio of thyroid hormones FT3/FT4 in table 1 is omitted ? the tests used need to be standard or proven by an international society or committe.

Reviewer #3: (No Response)

Reviewer #4: - It appears that a number of reviewers have already and thoroughly reviewed this manuscript, thus, I am not going to repeat what most of them observed/commented on, particularly Reviewer 1 whose review was thorough and insightful. I am only commenting on points that were not raised by other reviewers, or those which I think the authors’ response was not satisfactory.

- Reviewer 1 had issues interpreting ORs, to which the authors responded by modifying the wording. I still think the wording is confusing: for example, in the abstract the authors wrote:

- “Univariate analysis demonstrated that lower FT3 levels (OR = 0.85, 95% CI:28 0.80-0.91, p< 0.0001) and a reduced FT3/FT4 ratio (OR = 0.80, 95% CI: 0.69-0.92, p29= 0.0018) were significantly associated with an increased risk of iSAP.” And then in the next sentence they wrote:

- “After adjusting for confounders, multivariate analysis revealed that a higher FT3/FT4 ratio was independently associated with a lower iSAP risk (Q3 vs. Q1: OR= 0.40, 95% CI:32 0.26-0.62, p < 0.0001; Q4 vs. Q1: OR = 0.31, 95% CI: 0.19-0.48, p < 0.0001).”

- This is confusing: the first sentence implies the lower FT3 had OR<1 which was interpreted as “increased risk of iSAP”(?!), but the second sentence implies that higher FT3 (Q3 vs Q1 and Q4 vs Q1) were associated with, again, OR<1, but this time interpreted as “lower iSAP risk”. Later in main text, lines 252-255, the authors wrote: “Regarding thyroid hormone parameters, lower FT3 levels were associated with increased odds of developing iSAP (OR = 0.85, 95% CI: 0.80–253 0.91, p < 0.0001), as was a lower FT3/FT4 ratio (OR = 0.80, 95% CI: 0.69–0.92, p =254 0.0018)”. Similar to the first sentence in abstract above, this is incorrect: An increased odds of iSAP cannot be an OR of 0.85; This is an error in wording, not a counter-intuitive finding as the author suggested in one of their responses to reviewers’ comments. Results of univariate and multivariate analyses should be consistent in terms of what the ORs represent. As Reviewer 1 commented, an OR of <1 is generally perceived as lower risk, which is indeed reflected in the second sentence above in the authors’ own wording. I suggest the wording to be corrected in the first sentence, and elsewhere as needed, so that the reader is clear about the interpretation of these results. Alternatively, the authors could have considered re-defining the reference group for T3 levels, so that Q3 (the highest level) is the reference, and then compare to it Q2 and Q1 (lowest levels); had this been done, the OR would have reflected the increased risk in iSAP, with OR >1, which is more intuitive. In my view, the wording in the second sentence above is clear and correct, so I don’t see a need to re-do the analysis. Indeed, “higher T3 associated with lower risk of iSAP” is the equivalent (flip side if you well) of lower T3 associated with higher risk of iSAP.

- I think the use of case-control design-suggestive wording, such as referring to “cases” as those with iSAP and comparing then to “controls” who had no iSAP. This does not align with conventional epidemiologic wisdom. Since all participants had AIS, and the comparison is between those who did and did not develop iSAP, the analysis is essentially cross-sectional within a retrospective AIS cohort, as exposure was measured at a single point that preceded iSAP. By conditioning on AIS, the authors already restricted their base population and precludes conducting a methodologically sound case-control study. I therefore do not think the use of “controls” is sound in this manuscript. Referring, or even suggesting, to this as a case-control study risks confusion about control selection and the interpretation of odds ratios.

- How did the authors establish the criterion “Patients who had infectious diseases or fever within 2 weeks prior to admission and those who used antibiotics within 1 week before admission were excluded”? Do they have access to regional/state level health records, or just relied on their own centre’s records? I am concerned the CRP level in the iSAP group is near to 7 (seven) times that of the non-iSAP group, this is shortly after admission so I am wondering whether the former group was truly infection/pneumonia-free upon admission. The markedly elevated CRP levels observed at admission in the iSAP group (median 13.6 vs. 2.0 mg/L) raise concerns about potential misclassification of the outcome. It is plausible that some patients labeled as developing iSAP during hospitalization were already infected at admission, which would render the temporal relationship between thyroid hormone levels and infection ambiguous. The significantly higher prevalence of COPD in the iSAP group (15.7% vs. 4.0%) further supports the concern that these patients were at a fundamentally different baseline risk for infection and inflammation. While COPD was included as a covariate in the fully adjusted model, this binary adjustment does not adequately account for disease severity or its systemic inflammatory impact. More importantly, statistical adjustment cannot resolve the misclassification of prevalent pneumonia as incident iSAP, nor can it correct for potential reverse causality (i.e., low FT3 as a marker of acute illness rather than a causal factor). These limitations undermine the validity of the observed associations and should be acknowledged explicitly in the discussion. More importantly, rather than implying a protective or harmful effect, the observed association between FT3 and the odds of iSAP should be interpreted as hypothesis-generating, pending confirmation in longitudinal studies, as Reviewer 1 recommended.

- The authors wrote in their methods “The data for this study were derived from a previously established retrospective cohort study on the prognosis of ischemic stroke.” But they do not cite any studies. This is important to clarify whether the same results were published previously, in full or in part, and to clarify the added value of this submitted update.

- Most other reviewers commented on the need for language proofing, yet, still I see issues remaining. For example, line 114: “(AF)and “: missing space.

7. PLOS authors have the option to publish the peer review history of their article (what does this mean? ). If published, this will include your full peer review and any attached files.

**Do you want your identity to be public for this peer review?** For information about this choice, including consent withdrawal, please see our Privacy Policy .

Reviewer #2: No

Reviewer #3: No

Reviewer #4: **Yes: ** Omar Okasha, MD, MPH

---

## [Author Response · Author response to Decision Letter 2]

29 Jul 2025

Dear Editor and Reviewers,

We sincerely thank the editor and reviewers for their thoughtful and thorough evaluation of our manuscript entitled “Association Between Thyroid Hormone Sensitivity and Ischemic Stroke‑Associated Pneumonia: The Role of FT3/FT4 Ratio” (Manuscript ID: PONE‑D‑25‑07372R1). Their constructive comments greatly strengthened the manuscript’s clarity, methodological rigor, and overall quality.

In response, we carefully revised the manuscript and prepared detailed point‑by‑point responses. Key revisions include clarifying methodological procedures (e.g., matching methods and biomarker measurements), improving terminology and OR interpretation, adding supplementary data (e.g., balance diagnostics and updated tables), and expanding the discussion of study limitations. These revisions do not alter the study’s primary conclusions. They instead enhance its transparency and interpretability.

Below, we provide detailed responses to each comment from the reviewers, with corresponding revisions indicated in the manuscript (line numbers refer to the revised version).

Reviewer #2 – General Comment

We thank Reviewer #2 for raising detailed methodological points, including sample size imbalance, eGFR calculation and reference ranges, biomarker sample type, FT3/FT4 ratio reporting in Table 1, and standardization of testing methods. We have carefully revised the manuscript to address each of these points in detail, as summarized below.

Reviewer #2 Comment (Original)

The paper discusses an important point regarding thyroid hormones in stroke pateints, but the methods used to address the problem need to be improved, for example the difference between the control and study group number is large (iSAP group (n = 376) and the control group (n = 1,391). The eGFR values need to be repeated, what are the normal ranges of eGFR? Did you use serum or plasma for biomarker measurements? The ratio of thyroid hormones FT3/FT4 in table 1 is omitted? the tests used need to be standard or proven by an international society or committe.

Response:

Your comments highlighted important methodological and statistical issues requiring clarification, which led us to refine these sections. Please see our point-by-point responses below:

1. Large difference in sample sizes between the non‑iSAP and iSAP groups

Response: This comment drew our attention to the sample size imbalance between groups and prompted us to address it by combining propensity score matching (PSM) with Genetic Matching (GenMatch) to minimize baseline covariate imbalance and improve comparability.

Revision details: Matching was performed in a 1:1 ratio using nearest‑neighbor methods with replacement. This allowed certain iSAP patients to be matched multiple times to achieve optimal covariate balance and is widely used in observational studies when group sizes differ substantially.

New results: The final matched cohort included 2,926 patients (1,463 in each group). Supplementary Table 1 and Supplementary Figure 1 show that the absolute standardized mean differences (SMDs) for most covariates were reduced to below 0.1, indicating excellent matching quality.

Location in manuscript: Additional methodological details are provided in the revised Methods (Lines 199–208), and the balance analysis results are now presented in the Results (Lines 257–266).

Following matching, we performed multivariate logistic regression in the matched cohort. The results are shown in Table 4 (Line 376). The FT3/FT4 ratio was significantly associated with iSAP risk (Q2 vs. Q1: OR = 0.69, 95% CI: 0.56–0.86, p = 0.0011; Q4 vs. Q1: OR = 0.72, 95% CI: 0.57–0.91, p = 0.0054). TT4RI and TFQI‑FT4 showed positive associations, while TFQI‑FT3 remained significant in the fully adjusted model (Results, Lines 353–357).

We noted that TSHI and TFQI-FT3 exhibited attenuated or reversed associations after matching. This likely reflects:

(1) Improved balance in sample structure reducing residual confounding;

(2) Nonlinear interactions between TFQI indices and unmatched variables manifesting as directional shifts in extreme quartiles, a common phenomenon in observational studies.

2. Calculation method and reference range for eGFR

Response: We clarified in the revised manuscript that eGFR was calculated from serum creatinine (Scr) using the CKD-EPI formula.

Location in manuscript: Methods (Lines 145–146).

Reference range: The normal range is 90–120 mL/min/1.73 m².

3. Sample type used for biomarker measurements

Response: We clarified in the revised manuscript that serum creatinine was measured using serum samples.

Location in manuscript: Methods (Lines 139–140).

4. Omission of FT3/FT4 ratio in Table 1

Response: We corrected the omission and added FT3/FT4 ratio values for the control group to ensure completeness of baseline characteristics:

non-iSAP group: 0.31 (0.26–0.38)

iSAP group: 0.25 (0.21–0.30)

Location in manuscript: Revised Table 1 (Line 267).

5. Standardization of testing methods

Response: We clarified that all thyroid hormone indices were measured using electrochemiluminescence immunoassay (ECLIA) on the Roche Cobas analyzer, a method widely endorsed by international endocrine societies.

Location in manuscript: Methods (Lines 142–144).

Summary for Reviewer #2:

These revisions comprehensively address the reviewer’s concerns regarding methodological rigor and data clarity, while preserving the overall conclusions of the study.

Reviewer #4 – General Comment

We thank Reviewer #4 for insightful feedback highlighting key areas for improvement, such as OR interpretation, study design terminology, potential misclassification and COPD‑related confounding, data provenance, and minor language issues.

Reviewer #4 Comment (Original)

It appears that a number of reviewers have already and thoroughly reviewed this manuscript, thus, I am not going to repeat what most of them observed/commented on, particularly Reviewer 1 whose review was thorough and insightful. I am only commenting on points that were not raised by other reviewers, or those which I think the authors’ response was not satisfactory.

Comment 1: Odds ratio (OR) interpretation and wording

Reviewer 1 had issues interpreting ORs, to which the authors responded by modifying the wording. I still think the wording is confusing: for example, in the abstract the authors wrote:

- “Univariate analysis demonstrated that lower FT3 levels (OR = 0.85, 95% CI:28 0.80-0.91, p< 0.0001) and a reduced FT3/FT4 ratio (OR = 0.80, 95% CI: 0.69-0.92, p29= 0.0018) were significantly associated with an increased risk of iSAP.” And then in the next sentence they wrote:

- “After adjusting for confounders, multivariate analysis revealed that a higher FT3/FT4 ratio was independently associated with a lower iSAP risk (Q3 vs. Q1: OR= 0.40, 95% CI:32 0.26-0.62, p < 0.0001; Q4 vs. Q1: OR = 0.31, 95% CI: 0.19-0.48, p < 0.0001).”

This is confusing: the first sentence implies the lower FT3 had OR<1 which was interpreted as “increased risk of iSAP”(?!), but the second sentence implies that higher FT3 (Q3 vs Q1 and Q4 vs Q1) were associated with, again, OR<1, but this time interpreted as “lower iSAP risk”. Later in main text, lines 252-255, the authors wrote: “Regarding thyroid hormone parameters, lower FT3 levels were associated with increased odds of developing iSAP (OR = 0.85, 95% CI: 0.80–253 0.91, p < 0.0001), as was a lower FT3/FT4 ratio (OR = 0.80, 95% CI: 0.69–0.92, p =254 0.0018)”. Similar to the first sentence in abstract above, this is incorrect: An increased odds of iSAP cannot be an OR of 0.85; This is an error in wording, not a counter-intuitive finding as the author suggested in one of their responses to reviewers’ comments. Results of univariate and multivariate analyses should be consistent in terms of what the ORs represent. As Reviewer 1 commented, an OR of <1 is generally perceived as lower risk, which is indeed reflected in the second sentence above in the authors’ own wording. I suggest the wording to be corrected in the first sentence, and elsewhere as needed, so that the reader is clear about the interpretation of these results. Alternatively, the authors could have considered re-defining the reference group for T3 levels, so that Q3 (the highest level) is the reference, and then compare to it Q2 and Q1 (lowest levels); had this been done, the OR would have reflected the increased risk in iSAP, with OR >1, which is more intuitive. In my view, the wording in the second sentence above is clear and correct, so I don’t see a need to re-do the analysis. Indeed, “higher T3 associated with lower risk of iSAP” is the equivalent (flip side if you well) of lower T3 associated with higher risk of iSAP.

Response:

We appreciate the reviewer’s observation regarding this inconsistency. After reviewing both the abstract and results sections, we recognized that our previous wording may have caused confusion for readers. We have revised all relevant sentences so that OR < 1 is consistently interpreted as indicating a protective association. Specifically:

In the Abstract (Lines 28–30), we now describe higher FT3 levels and higher FT3/FT4 ratio as being associated with lower iSAP risk.

In the Results section (Lines 325–327), similar adjustments were made to maintain consistency.

These changes ensure that the statistical interpretation is accurate and presented in a way that is straightforward for readers to follow.

Comment 2: Use of “case-control” terminology

I think the use of case-control design-suggestive wording, such as referring to “cases” as those with iSAP and comparing then to “controls” who had no iSAP. This does not align with conventional epidemiologic wisdom. Since all participants had AIS, and the comparison is between those who did and did not develop iSAP, the analysis is essentially cross-sectional within a retrospective AIS cohort, as exposure was measured at a single point that preceded iSAP. By conditioning on AIS, the authors already restricted their base population and precludes conducting a methodologically sound case-control study. I therefore do not think the use of “controls” is sound in this manuscript. Referring, or even suggesting, to this as a case-control study risks confusion about control selection and the interpretation of odds ratios.

Response:

We appreciate the reviewer’s concern about potential confusion from case‑control terminology. To prevent this, we replaced the term “control” with “non‑iSAP” throughout the manuscript and tables and clarified this change in the Methods section.

In the Methods (Lines 96–100), we clarified that thyroid hormone indices were measured at admission and iSAP was assessed during hospitalization, underscoring the temporal relationship.

We also explicitly describe the study as a retrospective cohort rather than case-control design.

Comment 3: Potential misclassification of iSAP and baseline infection status / COPD confounding / hypothesis-generating framing

How did the authors establish the criterion “Patients who had infectious diseases or fever within 2 weeks prior to admission and those who used antibiotics within 1 week before admission were excluded”? Do they have access to regional/state level health records, or just relied on their own centre’s records? I am concerned the CRP level in the iSAP group is near to 7 (seven) times that of the non-iSAP group, this is shortly after admission so I am wondering whether the former group was truly infection/pneumonia-free upon admission. The markedly elevated CRP levels observed at admission in the iSAP group (median 13.6 vs. 2.0 mg/L) raise concerns about potential misclassification of the outcome. It is plausible that some patients labeled as developing iSAP during hospitalization were already infected at admission, which would render the temporal relationship between thyroid hormone levels and infection ambiguous. The significantly higher prevalence of COPD in the iSAP group (15.7% vs. 4.0%) further supports the concern that these patients were at a fundamentally different baseline risk for infection and inflammation. While COPD was included as a covariate in the fully adjusted model, this binary adjustment does not adequately account for disease severity or its systemic inflammatory impact. More importantly, statistical adjustment cannot resolve the misclassification of prevalent pneumonia as incident iSAP, nor can it correct for potential reverse causality (i.e., low FT3 as a marker of acute illness rather than a causal factor). These limitations undermine the validity of the observed associations and should be acknowledged explicitly in the discussion. More importantly, rather than implying a protective or harmful effect, the observed association between FT3 and the odds of iSAP should be interpreted as hypothesis-generating, pending confirmation in longitudinal studies, as Reviewer 1 recommended.

Reviewer Comment Part 1: Potential misclassification of iSAP and baseline infection status

“How did the authors establish the criterion… the markedly elevated CRP levels… raise concerns about potential misclassification…”

Response:

We implemented three specific measures to address these concerns:

1. Misclassification of iSAP

The exclusion of pre-existing infections relied on our hospital’s integrated EMR, which includes inpatient/outpatient data and provincial-level visit records. This system enables clinicians to review prior diagnoses and treatments.

We added a description in the Methods (Lines 91–96) clarifying this process and, in the Definitions section (Lines 149–152), specified that all patients underwent chest CT and infection screening at admission to ensure iSAP was hospital-acquired.

SAP diagnoses followed the Pneumonia in Stroke Consensus Group criteria and were independently confirmed by two neurologists, and respiratory specialists were consulted when necessary.

2. COPD confounding

We clarified in the Methods (Lines 128–130) that COPD diagnoses were based on confirmed medical records, rather than patient self‑reports.

In the Discussion – Limitations (Lines 535–539), we acknowledged residual confounding due to lack of COPD severity data.

3. Hypothesis-generating framing

We revised language across the manuscript to avoid implying causality.

In the Discussion – Limitations (Lines 556–561), we added:

“However, given the retrospective design, the observed associations between thyroid hormone sensitivity indices and iSAP should be interpreted as hypothesis-generating rather than implying causality. In particular, the inverse relationship between FT3 levels and iSAP risk may reflect a marker of disease severity rather than a causal factor. Therefore, longitudinal cohort studies are warranted to verify the temporal and mechanistic links.”

Similar wording changes were applied to the Abstract (Lines 30, 32, 34–35, 40–43), Discussion (Lines 425–427, 440–442, 471–472, 495–497) and Conclusion (Lines 563–568) for consistency.

Comment 4: Data provenance and potential duplication

The authors wrote in their methods “The data for this study were derived from a previously established retrospective cohort study on the prognosis of ischemic stroke.” But they do not cite any studies. This is important to clarify whether the same results were published previously, in full or in part, and to clarify the added value of this submitted update.

Response:

The data originate from a previously established stroke cohort, while the present analysis is novel, focusing on thyroid hormone sensitivity indices (FT3/FT4, TT4RI, TSHI, TFQI) and related hypotheses that have not been previously reported. The statistical methods used in this work are also original to this study, and we clarified this information in the Methods (Lines 81–86).

Comment 5: Language and typographical issues

Most other reviewers commented on the need for language proofing, yet, still I see issues remaining. For example, line 114: “(AF)and “: missing space.

Response:

We have conducted a thorough proofreading of the entire manuscript and corrected all typographical and formatting issues. This includes the missing space at line 114.

Summary for Reviewer #4

Collectively, these revisions clarify OR interpretation, re

---

## [Decision Letter · Decision Letter 2]

8 Sep 2025

Association Between Thyroid Hormone Sensitivity and Ischemic Stroke-Associated Pneumonia: The Role of FT3/FT4 Ratio

PONE-D-25-07372R2

Dear Dr. Rao,

We’re pleased to inform you that your manuscript has been judged scientifically suitable for publication and will be formally accepted for publication once it meets all outstanding technical requirements.

Kind regards,

Ennio Polilli

Academic Editor

PLOS ONE

Additional Editor Comments (optional):

Reviewer #4:

Reviewer #5:

Reviewers' comments:

Reviewer's Responses to Questions

**Comments to the Author**

1. If the authors have adequately addressed your comments raised in a previous round of review and you feel that this manuscript is now acceptable for publication, you may indicate that here to bypass the “Comments to the Author” section, enter your conflict of interest statement in the “Confidential to Editor” section, and submit your "Accept" recommendation.

Reviewer #4: All comments have been addressed

Reviewer #5: (No Response)

2. Is the manuscript technically sound, and do the data support the conclusions?

Reviewer #4: Yes

Reviewer #5: Yes

3. Has the statistical analysis been performed appropriately and rigorously? 

Reviewer #4: Yes

Reviewer #5: Yes

4. Have the authors made all data underlying the findings in their manuscript fully available?

Reviewer #4: Yes

Reviewer #5: Yes

5. Is the manuscript presented in an intelligible fashion and written in standard English?

Reviewer #4: Yes

Reviewer #5: Yes

6. Review Comments to the Author

Reviewer #4: The authors addressed all of my comments adequately and clearly. I thank them for their patience and I hope their important manuscript is published soon.

Reviewer #5: General comments:

This is a large retrospective study concerning the possible usefulness of free triiodothyronine (FT3), free thyroxine (FT4), FT3/FT4 ratio, thyroid-stimulating hormone index (TSHI), Thyrotroph T4 Resistance Index (TT4RI), and Thyroid Feedback Quantile-based Indices (TFQI-FT3, TFQI-FT4) in 1,767 euthyroid patients with ischemic stroke as predictors for the risk of ischemic stroke-associated pneumonia (iSAP). The study was well planned and conducted. The statistical analysis was properly done. After adjusting for confounders, multivariate analysis revealed that a higher FT3/FT4 ratio was inversely associated with iSAP occurrence. Elevated TFQI-FT3 levels also showed a significant inverse association with iSAP occurrence. Moreover, receiver operating characteristic (ROC) curve analysis demonstrated that the FT3/FT4 ratio and the Age, Atrial fibrillation, Dysphagia, Sex, Stroke severity (A2DS2) score exhibited moderate predictive accuracy for iSAP. The authors concluded that in euthyroid patients with ischemic stroke, a lower FT3/FT4 ratio and reduced TFQI-FT3 levels appear to be linked to higher odds of iSAP and suggest their use as predictive markers, encouraging their further validation by the enactment of prospective studies.

This manuscript has already undergone two rounds of revision by expert reviewers, which prompted the authors to make several major changes and integrations throughout the text.

This is a re-exploitation from a novel perspective of a dataset that already supported prior publication focusing on different aspects. Yet, the specific issue raised by this article is of interest and unprecedented.

The Methods section provides sufficient data to understand the conception of the study, the patient selection criteria, the data collection process, the diagnostic clinical and clinical biochemical workflow, and the statistical methodology adopted. All were sufficiently appropriate.

The Results are clearly described. The Tables are informative. The footnotes to Tables 2-4 can be lightened by referring to the same ones that appear at the bottom of Table 1 (see the particular comments below).

The Discussion is consistent with the data presented and offers useful implications and food for thought.

The Conclusions are sound.

The set of bibliographic references provides adequate support to the article.

Overall, it is my opinion that this manuscript deserves publication.

Particular comments:

- Line 38 (abstract): Please write A2DS2 in full followed by the abbreviation in brackets

- Please lighten the footnotes to Tables 2, 3, and 4, by referring to the corresponding ones that appear at the bottom of Table 1.

7. PLOS authors have the option to publish the peer review history of their article (what does this mean? ). If published, this will include your full peer review and any attached files.

**Do you want your identity to be public for this peer review?** For information about this choice, including consent withdrawal, please see our Privacy Policy .

Reviewer #4: **Yes: ** Omar Okasha, MD, MPH

Reviewer #5: **Yes: ** Fabrizio Gentile

---

## [Editor Report · Acceptance letter]

PONE-D-25-07372R2

PLOS ONE

Dear Dr. Rao,

I'm pleased to inform you that your manuscript has been deemed suitable for publication in PLOS ONE. Congratulations! Your manuscript is now being handed over to our production team.

Kind regards,

on behalf of

Dr. Ennio Polilli

Academic Editor

PLOS ONE